# TADA: TIMESTEP-AWARE DATA AUGMENTATION FOR DIFFUSION MODELS

## ABSTRACT

Data augmentation is a popular technique to improve the generalization performance of neural networks, particularly when dealing with limited data. However, simply applying augmentation techniques to generative models can lead to a distribution shift problem, producing unintended augmented-like output samples. While this issue has been actively studied in generative adversarial networks (GANs), little attention has been paid to diffusion models despite their widespread use. In this paper, we conduct the first comprehensive study of data augmentation for diffusion models, primarily investigating the relationship between distribution shifts and data augmentation. Our study reveals that distribution shifts in diffusion models originate exclusively from specific timestep intervals, rather than from the entire timesteps. Based on these findings, we introduce a simple yet effective data augmentation strategy that flexibly adjusts the augmentation strength depending on timesteps. Experiments on diverse diffusion model settings (e.g., noise schedule, model size, and sampling steps), datasets, and a training setup (e.g., training from scratch or transfer learning) show that our approach is applicable across different design choices, with minimal adjustments to the data processing pipeline. We expect that our data augmentation method can benefit various diffusion model designs and tasks across a wide scope of applications. We will make our code publicly available.

## 1 INTRODUCTION

Diffusion models (Sohl-Dickstein et al., 2015) have emerged as a *de facto* standard method for various tasks, including unconditional image synthesis (Ho et al., 2020; Nichol & Dhariwal, 2021), text-to-image synthesis (Rombach et al., 2022; Saharia et al., 2022), image restoration (Kawar et al., 2022), and image editing (Couairon et al., 2023; Kim et al., 2023). Recent diffusion models have demonstrated that their generation quality is on a par with those of generative adversarial networks (GANs) (Goodfellow et al., 2014) while offering the advantages of high diversity and stable training (Dhariwal & Nichol, 2021; Song et al., 2021). However, diffusion models depend on a substantial amount of training data, and they have not outperformed GANs in small datasets (e.g., AFHQ (Choi et al., 2020)). This limits the use of diffusion models in various applications, such as medical imaging and industrial design, where the amount of available data is restricted.

Data augmentation, a widely acknowledged approach to addressing this challenge, increases the number of training data by generating new data instances from existing data through various transformations (Shorten & Khoshgoftaar, 2019). While data augmentation is a common training technique in discriminative models (Cubuk et al., 2018; 2020), its application to generative models presents unique challenges. When generative models are trained with augmented data, they often produce augmented-like transformed samples caused by *distribution shift* (Zhao et al., 2020). This shift leads to the generation of unintended out-of-distribution samples (Jun et al., 2020; Zhao et al., 2020). Distribution shift has been actively discussed in the GANs literature, and many studies were conducted to address this problem (Karras et al., 2020; Zhao et al., 2020; Jiang et al., 2021).

Nevertheless, data augmentation for diffusion models remains less explored in the research community. To the best of our knowledge, only Karras et al. (2022) have utilized data augmentation to mitigate overfitting in diffusion models, employing a method similar to Jun et al. (2020) proposed for likelihood-based models. Specifically, they condition the model with the augmentation type dur-

ing training. This enables the model to generate in-distribution samples when the condition is set to an identity function. However, integrating conditional input requires additional complexity, and the performance of diffusion models with conditional input can vary depending on the design choice (Appendix H).

In this study, we propose a data augmentation strategy that seamlessly integrates into the data pre-processing pipeline, requiring no extra modifications to the training procedure. Our inspiration comes from recent studies that diffusion models play a different role depending on the timesteps (i.e., noise levels) (Choi et al., 2022; Balaji et al., 2022; Feng et al., 2023). We begin by carefully investigating the impact of naive data augmentation on the reverse process of diffusion models. Specifically, we measure the prediction error along the sampling procedure. Through an in-depth analysis, we identify vulnerable timesteps where data augmentation has a substantial effect on performance degradation. We further unveil that specific timesteps contribute to alterations in the sampling trajectory, arriving at either out-of-distribution (i.e., unintended transformed outputs) or in-distribution sample generation. Based upon these findings, we propose Timestep-Aware Data Augmentation (TADA), which adaptively adjusts the strength of augmentation along timesteps.

Our augmentation pipeline, denoted as $\mathcal{T}(\boldsymbol{x}_t, w_t)$, modulates the augmentation strength $w_t$ based on the timestep $t$. In specific, we train the model with strongly augmented training samples (large $w_t$) when input contains a high level of noise. During the vulnerable timesteps, we apply small perturbations ($w_t$ near zero) to training samples. Finally, we increase the strength again when inputs become nearly clean images with a low-level noise. This approach allows diffusion models to benefit from data augmentation while preventing a potential distribution shift in the generated output.

Our method is simple, effective, and flexible. With extensive experiments, we confirm that TADA is applicable regardless of diverse diffusion model settings (e.g., noise schedule or sampling steps), high-resolution image generation, and various types of data augmentations. Our method achieves high-quality results when combined with transfer learning, a popular training technique for limited data scenarios. Our main contributions are summarized as follows:

- We present the first comprehensive study of data augmentation for diffusion models.

- We find that data augmentation plays a critical role in specific timesteps, which leads to distribution shifts.

- We propose a timestep-aware data augmentation technique, dubbed TADA, that requires minimal engineering and is applicable across various diffusion model settings.

- We empower diffusion models to achieve high performance with limited data, to be effective for both training from scratch or transfer learning.

## 2 RELATED WORK

**Timestep-wise role of diffusion models.**   Recent studies have interpreted training diffusion models as a form of mixture-of-experts, where each noise-level (or timestep) corresponds to a different task (Go et al., 2023). Choi et al. (2022) propose a weighting scheme for diffusion models by categorizing the forward process into multiple stages based on the perceptual noise level. Similarly, Hang et al. (2023) apply different weights according to task difficulty for more efficient training. Deja et al. (2022) divide the model into the generator and the denoiser, based on their observations that the final steps in the reverse diffusion process are data-agnostic. Balaji et al. (2022) and Feng et al. (2023) utilize ensembles of diffusion models for text-to-image synthesis, while Lee et al. (2023) employ distinct architectures for each expert to balance frequency components. These studies collectively leverage the role of timesteps in diffusion models and develop a mixture of multiple experts, where each expert specializes in the role of a specific timestep. Our work shares a similar intuition with these studies. We explore the role of timesteps in diffusion models to devise a novel data augmentation strategy tailored for diffusion models.

**Data augmentation for generative models.**   There have been many studies to solve the distribution shift problem in the literature of GANs. Zhao et al. (2020) apply differentiable augmentation to both real and fake images during training to the generator and the discriminator. Concurrent work by Karras et al. (2020) uses a similar approach while they demonstrated the effectiveness, particularly in limited data settings. Meanwhile, Jun et al. (2020) introduce a distribution augmentation

(DistAug) for likelihood-based models (especially auto-regressive models) by inducing the model to learn the conditional data density, taking the transformation parameters as a condition. Karras et al. (2022) utilize a method similar to DistAug in order to mitigate overfitting in diffusion models. In contrast to previous methods, our approach does not require extra effort in formulating conditional inputs. Instead, we train the diffusion model by simply adjusting the degree of augmentation at the data-processing stage.

## 3 METHOD

In this section, we investigate the impact of data augmentation on the sampling process of diffusion models. Our analysis reveals that specific timesteps (or noise levels) play a pivotal role in determining the sampling trajectories, either toward the original data distribution or the augmented data distribution. Built upon this analysis, we introduce a simple and effective data augmentation method: Timestep-Aware Data Augmentation (TADA).

### 3.1 PRELIMINARIES

**Diffusion models.** Consider a dataset of $n$ data points $\{\boldsymbol{x}_0^1, \dots, \boldsymbol{x}_0^n\}$ sampled from the distribution $q(\boldsymbol{x}_0)$. Diffusion models aim form a model $p_\theta(\boldsymbol{x}_0)$ that closely approximates the distribution $q(\boldsymbol{x}_0)$. Training diffusion models involves two key processes: the *forward process* where noisy data $\boldsymbol{x}_t$ is generated by adding standard Gaussian noise $\boldsymbol{z} \sim \mathcal{N}(\boldsymbol{0}, \boldsymbol{I})$ at timestep $t$, and the *reverse process* which aims to remove this noise to recover the original data.

In the forward process, $\boldsymbol{x}_t$ can be computed from each data point $\boldsymbol{x}_0$ as $\boldsymbol{x}_t = \alpha_t \boldsymbol{x}_0 + \sigma_t^2 \boldsymbol{z}$, where $\alpha_t > 0$ and $\sigma_t > 0$ are scalar-valued functions defined for $t \in [0, T]$. The model $\hat{\boldsymbol{\epsilon}}_\theta(\boldsymbol{x}_t, t)$ is trained to predict the added noise $\boldsymbol{z}$ at timestep $t$, by minimizing weighted mean squared error (Ho et al., 2020).

In the reverse process, $\boldsymbol{x}_{t-1}$ can be sampled from $p_\theta(\boldsymbol{x}_{t-1}|\boldsymbol{x}_t)$, starting from $\boldsymbol{x}_T \sim \mathcal{N}(\boldsymbol{0}, \boldsymbol{I})$. That is,

$$\hat{\boldsymbol{x}}_{t-1} = \alpha_{t-1}\hat{\boldsymbol{x}}_\theta(\boldsymbol{x}_t, t) + \sigma_{t-1}^2 \boldsymbol{z}, \tag{1}$$

where $\hat{\boldsymbol{x}}_\theta(\boldsymbol{x}_t, t) = (\boldsymbol{x}_t - \sigma_t^2 \hat{\boldsymbol{\epsilon}}_\theta(\boldsymbol{x}_t, t))/\alpha_t$.

Note that various options exist for $\alpha_t$ and $\sigma_t^2$, training objectives, and sampling methods. In our work, we adopt the approach proposed by Nichol & Dhariwal (2021), which is an enhanced version of the method introduced by Ho et al. (2020). Other choices have been explored in the literature, as mentioned in Song et al. (2020; 2021); Kingma et al. (2021); Karras et al. (2022).

**Signal-to-noise ratio (SNR).** Kingma et al. (2021) introduced the concept of signal-to-noise ratio, denoted as $\text{SNR}(t) = \alpha_t^2/\sigma_t^2$, which quantifies the noise level at each timestep $t$. Note that SNR is a monotonically decreasing function of $t$.

**Data augmentation.** Throughout this paper, we denote our data augmentation as $\mathcal{T}(\boldsymbol{x}_0, w)$. Here, $w \in [0, 1]$ represents the normalized hyper-parameter which controls the *strength* of augmentation. A large value of $w$ denotes strong augmentation, while a small value of $w$ indicates weak augmentation.

### 3.2 ANALYZING THE EFFECT OF DATA AUGMENTATION ON LEARNED REVERSE PROCESS

We conduct exploratory experiments to investigate the impact of data augmentation on the learned reverse process of diffusion models. We specifically compare the reverse process of diffusion models trained with and without augmentation, aiming to identify the timesteps where the model is susceptible to producing unintended output as a result of data augmentation.

**Toy experiment 1.** During the reverse process, a diffusion model predicts $\hat{\boldsymbol{x}}_0 = \hat{\boldsymbol{x}}_\theta(\boldsymbol{x}_t, t)$ at each timestep. We design a simple experiment by examining $\hat{\boldsymbol{x}}_0$ throughout the reverse process of a model trained with data augmentation. In essence, this can reveal when in the sampling process the diffusion model is prone to producing unintended output due to data augmentation.

Our analysis focuses on comparing the reverse process of two diffusion models: ($\hat{\boldsymbol{\epsilon}}_{base}$) a baseline solely trained with horizontal-flip and ($\hat{\boldsymbol{\epsilon}}_{aug}$) a model trained on the augmented data. Here, we

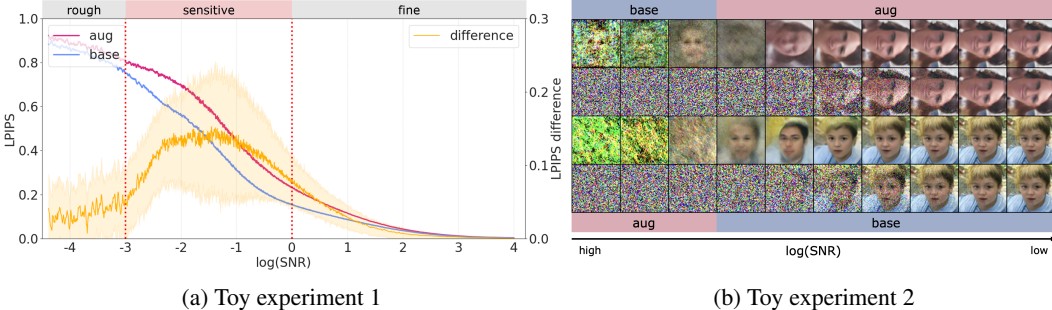

(a) Toy experiment 1            (b) Toy experiment 2

Figure 1: **Results of toy experiments**. (a) Blue and pink lines show the average LPIPS between $x_0$ and $\hat{x}_0$ produced by each model at every timestep, on 200 randomly selected input images from the FFHQ-5k dataset. Yellow line traces the LPIPS difference between the two lines with standard deviation (shaded). We plot based on a $\log(\text{SNR})$ scale to align with perceptual aspects of LPIPS. (b) Sampling trajectory of the same input image with added noise. Each row shows $x_t$s and $x_0$s for every $\log(\text{SNR})$ from -5 to 5, respectively. Samples in the first two rows corresponds to a case where a noisy sample is denoised by $\hat{\epsilon}_{base}$ and then denoised by $\hat{\epsilon}_{aug}$ from the middle. The last two rows illustrate the opposite case.

adopted the augmentation pipeline used in Karras et al. (2020). For each timestep $t$, we (1) randomly select $N$ samples $x_0 \sim q(x_0)$, (2) add noise to obtain $x_t$, (3) perform a single step of the reverse process to obtain $\hat{x}_0$ from each model, which we denote as $\hat{x}_{base}(x_t, t)$ and $\hat{x}_{aug}(x_t, t)$, and (4) measure the perceptual similarity (Zhang et al., 2018) between $\hat{x}_0$ and input image $x_0$: $\text{LPIPS}(\hat{x}_{base}(x_t, t), x_0)$ and $\text{LPIPS}(\hat{x}_{aug}(x_t, t), x_0)$.

In Figure 1a, we marked the difference between the two models with corresponding values reported on the right axis. The result shows that there is no clear difference between the two models during both the initial and final stages of the reverse process. During the initial stage, both models predict results far from $x_0$ because $x_t$ contains little information about $x_0$ (low SNR). In other words, when a high level of noise is added to both the original image $x_0$ and the augmented sample, these two images become perceptually indistinguishable (Choi et al., 2022). In the final timesteps, both models achieved nearly perfect predictions of the original image $x_0$. This can be interpreted as the role of the final timesteps being data-agnostic refinement, without modifying or introducing additional contextual information to the sample (Deja et al., 2022). Intriguingly, Choi et al. (2022) and Deja et al. (2022) have reported similar findings, while the exact timestep region is slightly different from our observations. In this regard, we denote initial timesteps as *rough* and final timesteps as *fine* from this section.

**Toy experiment 2.** Meanwhile, during the timesteps in the middle, the distinction between the two models becomes apparent, as in the green line of Figure 1a. To examine this further, we performed another experiment. We create the two different sampling processes by interchanging the reverse process between $\hat{\epsilon}_{aug}$ and $\hat{\epsilon}_{base}$ at the middle intervals: $\hat{\epsilon}_{base} \rightarrow \hat{\epsilon}_{aug}$ and $\hat{\epsilon}_{aug} \rightarrow \hat{\epsilon}_{base}$. The results of both sampling procedures are visualized in Figure 1b.

The first two rows of Figure 1b illustrate a case of $\hat{\epsilon}_{base} \rightarrow \hat{\epsilon}_{aug}$, where a sample initially follows the data distribution but ends up in the augmented data distribution. In other words, even if a sample follows the trajectory towards the data distribution by $\hat{\epsilon}_{base}$, it can be altered and result in an augmented-like output by $\hat{\epsilon}_{aug}$. On the contrary, the last two rows demonstrate a case where a sample initially following the trajectory of $\hat{\epsilon}_{aug}$ was adjusted by $\hat{\epsilon}_{base}$ during the middle timesteps, ultimately aligning with the data distribution. These two examples demonstrate that the middle timesteps in a diffusion model can alter the sampling trajectory, determining whether the final sample adheres to the original data distribution or the augmented one. We further performed this experiment by swapping the trajectory at the other timesteps. However, these modifications did not affect the generated data distribution (can be found in Appendix C.2). To this end, we denote the timesteps of the middle as *sensitive* timesteps.

In summary, the experiments in this section indicate that data augmentation has a significant impact on *sensitive* timesteps, while the *rough* and *fine* timesteps remain relatively unaffected. Note

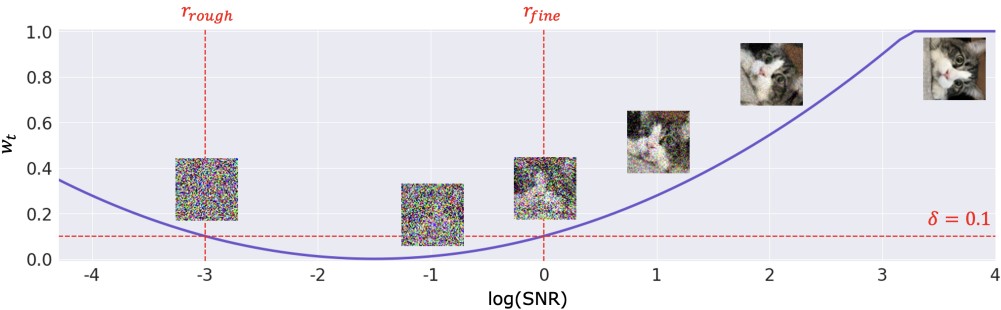

Figure 2: **Illustration of TADA**. Purple line indicates augmentation strength $w_t$ with our default parameters. Image illustrates an example of $\boldsymbol{x}_t$ at corresponding $\log(\mathrm{SNR}(t))$. Note that this showcases an instance of *rotation*. In practice, we apply different types of augmentations simultaneously.

that the choice of different settings does not change the trend of this experiment (please refer to Appendix C.1).

### 3.3 TIMESTEP-AWARE DATA AUGMENTATION FOR DIFFUSION MODELS

Based on the empirical study of the previous section, we propose to apply strong augmentation for both the *rough* and *fine* timesteps while decreasing the augmentation strength during the *sensitive* timesteps.

Our `Timestep-Aware Data Augmentation` (TADA) assigns different strengths of data augmentation according to its timesteps, adhering to the following rules:

- **Sensitive timesteps** ($t \in [t_{rough}, t_{fine}]$): Augmentations of weak strength are applied to the training data to ensure that the generated samples remain close to $p(\boldsymbol{x})$.
- **Rough** ($t \in (t_{rough}, T]$) **& Fine** ($t \in [0, t_{fine})$) **timesteps**: Strong augmentations are used to prevent overfitting and improve the generalization capability of diffusion models.

By following this design philosophy, our augmentation strategy denoted as $\mathcal{T}(\boldsymbol{x}_t, w_t)$ can be expressed in terms of the augmentation strength $w_t$:

$$w_t = \kappa(r_t - r_{rough})(r_t - r_{fine}) + \delta, \tag{2}$$

where $r_t = \log(\mathrm{SNR}(t))$, and $\delta$ denotes the maximum augmentation strength in the sensitive region. We clipped $w_t$ to ensure its values remained within the available augmentation strength range (i.e., $0 \le w_t \le 1$). We set $\delta = 0.1$ as we observed that sufficiently low $w_t$ does not cause distribution shifts (Appendix D.1). By default, we automatically compute $\kappa$ so that the value of $w_t$ becomes 0 before clipping occurs (Appendix D.2). Albeit the simplicity, our formulation in Equation 2 can represent various profiles for $w_t$, including a sharp transition at $r_{rough}$ and $r_{fine}$ or smooth transitions, depending on the choice of $\kappa$ (Figure 5). Herein, the optimal choice of $r_{rough}$ and $r_{fine}$ varies slightly depending on the datasets (Appendix E). However, for the sake of robustness, we set the same $r_{rough}$ and $r_{fine}$ across all our experiments. An intuitive illustration of Equation 2 is provided in Figure 2. Please refer to Section 4.1 for more details on our augmentation pipeline.

**SNR calibration.** Our method and toy experiments are developed upon experiments on $64 \times 64$ resolution. This can be easily extended to various resolutions. Recent studies (Hoogeboom et al., 2023; Chen, 2023) have highlighted that the perceptual noise level of diffusion models varies with resolution, necessitating an adjustment in SNR. As our approach was performed based on perceptual similarity, we adjusted the augmentation strength using the equation proposed by Hoogeboom et al. (2023),

$$\mathrm{SNR}_{calibrated}(t) = \mathrm{SNR}(t) \,/\, (d/64)^2, \tag{3}$$

where $d$ and $t$ denote the image resolution and timestep, respectively. This adjustment on different resolutions can be done by simply plugging this equation into $r_t$ of Equation 2, enabling general usage of our method and thereby reducing the need for extra experiments at different resolutions. More details on SNR calibration can be found in Appendix F.3.

Table 1: **Results on different subsets of FFHQ 64×64.** We measured FID and KID every 10,000 training iterations and reported the value achieved at the iteration with the best FID. Both FID and KID were measured using 10,000 samples generated using 250 sampling timesteps.

| Methods | 1k | | 2k | | 5k | | 10k | | 30k | |
|---|---|---|---|---|---|---|---|---|---|---|
| | FID ↓ | KID $(\times 10^3)$ ↓ | FID ↓ | KID $(\times 10^3)$ ↓ | FID ↓ | KID $(\times 10^3)$ ↓ | FID ↓ | KID $(\times 10^3)$ ↓ | FID ↓ | KID $(\times 10^3)$ ↓ |
| h-flip | 17.46 | 11.18 | 36.22 | 30.10 | 14.84 | 9.30 | 13.31 | 9.87 | 12.00 | 7.79 |
| AR | 22.98 | 17.11 | **16.33** | **9.58** | 12.95 | 8.45 | **11.34** | **7.79** | 10.82 | 7.56 |
| TADA | **14.37** | **8.91** | 18.83 | 14.39 | **12.92** | **7.74** | 12.19 | 8.06 | **10.39** | **6.80** |

## 4 EXPERIMENTS

### 4.1 SETUPS

To evaluate our method under a limited data setting, we use a subset of FFHQ (Karras et al., 2019), as done in previous studies (Karras et al., 2020; Hou et al., 2021). Furthermore, we performed evaluations on AFHQ-v2 (Choi et al., 2020; Karras et al., 2021).

To demonstrate the effectiveness of TADA, we compare it with two augmentation methods from previous works: 1) 50% horizontal flip (i.e., h-flip), which has been commonly employed in diffusion models (Ho et al., 2020), and 2) Augmentation regularization method (AR) used in Karras et al. (2022). Our implementation is based on ADM (Dhariwal & Nichol, 2021), a commonly used baseline in recent studies (Choi et al., 2021; 2022; Go et al., 2023). Unless specified otherwise, we trained all models using linear scheduling with $T = 1000$. More details, including model configurations, can be found in Appendix A.

**Evaluation metrics.** To assess the quality of generation, we use Fréchet Inception Distance (FID) (Heusel et al., 2018) and Kernel Inception Distance (KID) (Bińkowski et al., 2018), as used in previous studies (Karras et al., 2020; Choi et al., 2022). Recently, Parmar et al. (2022) observed that resizing functions have a critical impact on these metrics and suggested the use of *clean* resizing methods (i.e., PIL Bicubic). To address this concern, we report FID and KID using *clean* resizing methods. If not specified otherwise, we measured FID and KID using 10,000 samples.

**Augmentation pipeline.** Following EDM (Karras et al., 2022), we implement TADA using a subset of augmentations proposed by Karras et al. (2020). Additionally, we incorporate color transformations, which were not used in EDM. We simultaneously apply $n \sim \{1, \ldots, M\}$ number of augmentations with a probability $p$, where $M$ indicates the maximum number of augmentations that can be applied to a training sample. This controls the extent to which the augmented data distribution deviates from the original distribution. For the strength of each augmentation, we use the same minimum and maximum strengths as defined by Karras et al. (2020). We control the strength at each timestep with $w_t$, as defined in Equation 2. Throughout the experiments, we set $p = 0.8$ and $M = 2$, with horizontal flip applied with the probability of $0.5$, independent of other augmentations. See Appendix D.3 for further details.

### 4.2 BENEFIT OF TADA

In this section, we evaluate the effectiveness of TADA on the quality of generated images under limited data settings. We show that our method does not produce distribution-shifted samples, due to the adjustment of augmentation strength during *sensitive* timesteps. Furthermore, we show that our method mitigates the overfitting problem, thereby enhancing the diffusion model's performance on small datasets.

**Effectiveness on limited data.** To evaluate TADA on different dataset sizes, we train the model on the subsets of FFHQ. Table 1 shows the best FID and KID results, with a total training iteration is 100,000. On each subset, we evaluate 10,000 samples generated with 250 sampling steps. In all datasets, TADA performs better than h-flip with a noticeable gap. The performance of our method is comparable to AR, suggesting that our simple solution can perform as effectively as AR, without any additional conditioning modules. We keep the same $r_{rough}$ and $r_{fine}$ values across all experiments for robustness on various design choices. Please refer to Appendix E for the results on different $r_{rough}$ and $r_{fine}$.

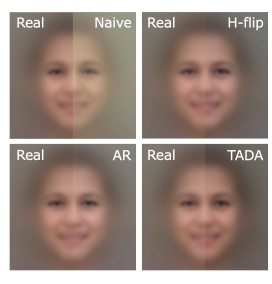

Figure 3: **Mean faces.**

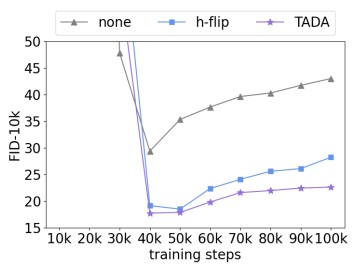

Figure 4: **Learning curves.**

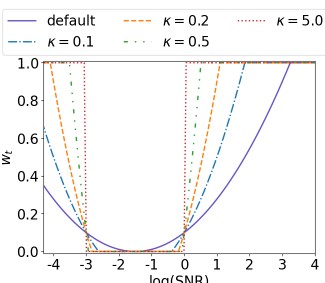

Figure 5: $\kappa$ **variation.**

Table 2: **Results for the 256×256 resolution.** We measured FID and KID at the last training iteration, as generating samples at 256×256 requires substantial computational resources. Both FID and KID were calculated using 10,000 samples generated over 50 sampling timesteps.

Table 3: **Transfer learning on 256×256 dataset.** All models were trained for 60,000 iterations on each domain of AFHQ-v2 using a model pre-trained on the FFHQ dataset.

| Methods | FFHQ 1k | | FFHQ 2k | | FFHQ 5k | |
|---|---|---|---|---|---|---|
| | FID ↓ | KID (×10³) ↓ | FID ↓ | KID (×10³) ↓ | FID ↓ | KID (×10³) ↓ |
| h-flip | 64.54 | 43.18 | 47.64 | 32.04 | 45.30 | 30.89 |
| AR | **39.73** | **20.56** | 40.45 | 25.61 | **38.50** | 29.24 |
| TADA | 41.29 | 22.53 | **35.75** | **23.17** | 39.15 | **28.53** |

| Dataset | AR | | TADA | |
|---|---|---|---|---|
| | FID ↓ | KID (×10³) ↓ | FID ↓ | KID (×10³) ↓ |
| Cat | 17.30 | 14.85 | **16.76** | **14.45** |
| Dog | 35.07 | 24.79 | **32.03** | **21.84** |
| Wild | 13.77 | 8.05 | **12.17** | **6.96** |

**Distribution shift.** To verify whether TADA generates in-distribution samples, we visualize mean faces, similar to Karras et al. (2020). Figure 3 shows the mean faces of 10,000 samples generated by each method, where 'Real' indicates the mean face of the 70,000 FFHQ dataset. With naive augmentation (trained with augmented samples in all timesteps), its mean face shows noticeable blurry eyes and huge color shifts compared to real images, evidence of distribution shift. Unlike naive augmentation, TADA produces a mean face with clear eyes.

Note that the colors of h-flip, AR, and TADA are slightly different from that of real images, where h-flip and TADA result in slightly darker images and AR results in slightly brighter images. We conjecture that this little color difference is due to an inherent problem of diffusion models, as mentioned by Choi et al. (2022). We further measure our effectiveness in addressing the distribution shift problem quantitatively, as further explained in Appendix F.1.

**Alleviating overfitting.** Previous studies (Karras et al., 2022; Jun et al., 2020) have demonstrated that data augmentation brings a regularization effect, thus mitigating overfitting. To investigate the impact on overfitting, we trained the baseline models without any augmentation (gray), with h-flip (blue), and compared the learning curve with our TADA (purple) on the FFHQ-5k dataset. Figure 4 depicts the FID scores at different training iterations. TADA shows its effectiveness in alleviating overfitting than h-flip.

## 4.3 GENERALIZATION

To demonstrate that TADA can be generalized to various design configurations in diffusion models, we conduct experiments with higher image resolutions, transfer learning scenarios, various noise scheduling, and different model sizes. Furthermore, we show our method is robust to different sampling timesteps.

**Higher resolution and transfer learning.** We tested our augmentation pipeline on 256×256 resolution, commonly used for comparing generative models (Karras et al., 2020; Rombach et al., 2022). Consistent with the results on lower resolution in Table 1, TADA achieves a clear improvement over h-flip and presents the competitiveness against AR without additional conditioning overhead in Table 2. Figure 6 shows the qualitative results of the model trained using TADA. Furthermore, we evaluated TADA in a transfer learning setting, frequently employed in a data-limited environment (Karras et al., 2020). Table 3 highlights that our method consistently outperforms AR with a noticeable margin across three subsets of the AHFQ-v2 dataset in a transfer learning scenario. This improvement is particularly meaningful because data augmentation with transfer learning is often a final solution for handling limited data. See Appendix I.1 for qualitative comparisons.

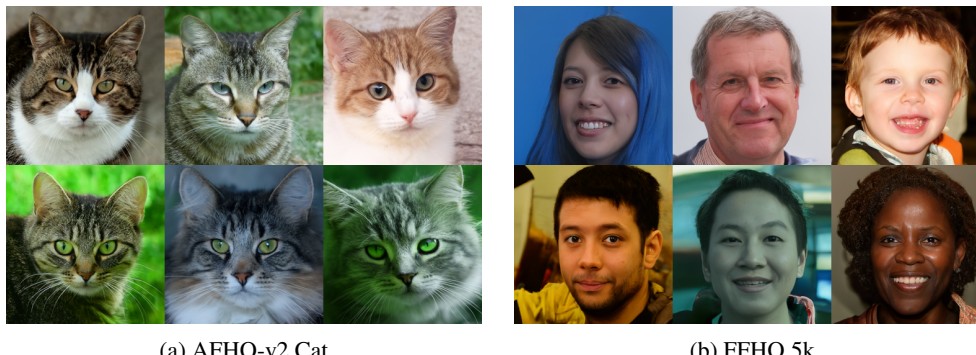

(a) AFHQ-v2 Cat       (b) FFHQ 5k

Figure 6: **Qualitative results.** We trained models *from scratch* at a 256×256 resolution using (a) AFHQ-v2 cat and (b) a 5k subset of FFHQ. We generated samples using 250 sampling timesteps.

Table 4: **Generalization.** We reported the FID value after 50,000 iteration of training. FID was measured using 10,000 samples generated with 250 timesteps, except for columns in the middle that used different sampling steps.

| | Noise scheduling | | Sampling steps | | | | Model size | | |
|---|---|---|---|---|---|---|---|---|---|
| | linear | cosine | 50 | 100 | 500 | 1000 | small (17M) | base (68M) | large (95M) |
| h-flip | 14.84 | 14.70 | 18.48 | 16.25 | 14.41 | 14.06 | **20.78** | 14.84 | 14.18 |
| TADA | **12.92** | **14.29** | **17.86** | **15.04** | **12.64** | **12.55** | 22.10 | **12.92** | **12.04** |

**Noise scheduling.** To test whether TADA works under different noise scheduling, we evaluate our method on both linear (Ho et al., 2020) and cosine (Nichol & Dhariwal, 2021) noise schedule. As shown in Table 4, TADA outperforms h-flip in both noise schedules. As we used the same $r_{rough}$ and $r_{fine}$ under different noise schedules, our method can be simply plugged in even under cosine noise scheduling, without any modification on the values of sensitive region boundaries.

**Sampling step.** Recent research has attempted to improve the inference efficiency by investigating further into the sampling procedure (Song et al., 2020; Nichol & Dhariwal, 2021) of diffusion models. Since data augmentation methods must coexist with other training and inference techniques, we evaluate our method across various sampling steps. Table 4 shows the performance trend of our TADA and h-flip at {50, 100, 500, 1000} sampling steps. Our TADA consistently outperforms h-flip, with the best FID achieved at 1000 steps, similar to previous studies (Choi et al., 2022).

**Model size.** We assessed the impact of TADA on different scales of the model. TADA shows a clear benefit with a large-scale model while less effective in a small-scale model. This outcome is expected because data augmentation is a critical recipe for enhancing the performance of large models.

### 4.4 ABLATION STUDY

To dissect the impact of each component within our TADA, we conducted three ablation studies. That is, we examine the impact of each time range under two different sampling steps (Table 5a, FID scores) and the choices of $M$ (Table 5b, FID and KID) and $\kappa$ (Table 5c, FID scores).

**Augmentation range.** In Table 5a, we exclusively apply $w_t$ at a specific timestep range while setting $w_t = 0$ for other ranges to obtain the FID scores for *rough*, *sensitive*, and *fine*. Considering the results of 'none' as the baseline performances (h-flip), we observe that the contributions from each range slightly vary across sampling steps: the *sensitive* range is the most influential in 50 sampling steps, while the *fine* range is the most impactful in 250 sampling steps. Nonetheless, we consistently observe TADA enjoys the benefit of augmentations from various ranges regardless of sampling steps, with a clear gain over the baseline.

$M$ **variation.** We test our TADA with different values of $M \in \{0, 1, 2, 3, 4\}$ and observe the resulting FID scores at 250 sampling steps in Table 5b. Compared to the baseline performance of

Table 5: **Ablation study.** All models are trained on FFHQ-5k (64x64) dataset. We measure FID and KID of 10k samples, generated with 250 timesteps. The best values are marked in **bold**, and the second-best values in each column are underlined.

<table>
<tr><td colspan="3" align="center">(a) Augmentation range.</td><td colspan="3" align="center">(b) $M$ variation.</td><td colspan="3" align="center">(c) $\kappa$ variation.</td></tr>
<tr><td>Range</td><td>50 steps</td><td>250 steps</td><td>M</td><td>FID ↓</td><td>KID $_{(\times 10^3)}$ ↓</td><td>$\kappa$</td><td>linear</td><td>cosine</td></tr>
<tr><td>none</td><td>18.48</td><td>14.84</td><td>0</td><td>14.84</td><td>9.30</td><td>0.1</td><td>13.19</td><td>13.22</td></tr>
<tr><td>rough</td><td>18.33</td><td>16.26</td><td>1</td><td>13.88</td><td>8.67</td><td>0.2</td><td>14.81</td><td>**11.37**</td></tr>
<tr><td>sensitive</td><td>**17.71**</td><td>14.65</td><td>2</td><td>**12.92**</td><td>**7.74**</td><td>0.5</td><td>14.46</td><td>12.22</td></tr>
<tr><td>fine</td><td>18.60</td><td>13.85</td><td>3</td><td>13.43</td><td>8.34</td><td>5.0</td><td>14.66</td><td>12.15</td></tr>
<tr><td>TADA</td><td>17.74</td><td>**12.92**</td><td>4</td><td>17.28</td><td>11.92</td><td>default</td><td>**12.92**</td><td>14.29</td></tr>
</table>

$M = 0$ (h-flip), we confirm that applying augmentations proves beneficial when $M$ is moderately small, such as 2. However, a large $M$ can deviate the training data distribution too far from the original training data, which may hinder model training. From this empirical study, we set $M = 2$ for all settings throughout this paper.

$\kappa$ **variation.** To evaluate the impact of $\kappa$ on TADA, we tested our method with varying $\kappa$ for both linear and cosine noise scheduling. As $\kappa$ varies, the augmentation strength varies accordingly (See Figure 5). Table 5c shows the FID scores by varying $\kappa$. Despite the fact that manually chosen $\kappa$ outperforms our default setting, it still consistently outperforms the h-flip method. The results demonstrate that our default value is a reasonable choice, but the performance of TADA can be improved with further hyper-parameter tuning.

## 5 DISCUSSION

**Comparison between TADA and augmentation regularization.** As shown in Section 4, TADA used for training from scratch achieved competitive results compared to augmentation regularization (AR) methods used in previous studies (Kingma et al., 2021; Karras et al., 2022). It should be noted that our TADA can further achieve improved performance by tuning thresholds per dataset by sacrificing robustness. Besides, our method is applied to the data processing pipeline without additional modules like AR. Finally, when our TADA is combined with transfer learning, it demonstrates clear gains in generation quality. Furthermore, while EDM did not find color transformations useful, we also take advantage of color transformations, which have been useful for TADA (Appendix 12).

**Limitation & future work.** Similar to other data augmentation methods for generative models, our approach cannot fully address the data scarcity issue, less effective than collecting more data. In addition, the augmentation set used in this paper was adapted from those commonly used in GANs, which may not be the optimal choice for diffusion models. Investigating the most effective augmentation combinations for diffusion models or designing a new augmentation transformation for diffusion models can be an interesting avenue for future research.

## 6 CONCLUSION

In this study, we conducted the first in-depth investigation on the relationships between data augmentation and distribution shift in diffusion model training. We discovered that data augmentation significantly influenced specific timestep intervals, giving rise to distribution shifts in diffusion models. Drawing from these observations, we introduced TADA, a method that adjusts data augmentation strength depending on the timestep. Our results demonstrate that even a very simple implementation of TADA consistently enhances the diffusion models across datasets, model configurations, and sampling steps. TADA effectively improves the generation performance of diffusion models in data-limited environments, addressing overfitting, and facilitating improved training from scratch and transfer learning. We expect our research to stimulate fresh avenues of exploration within the field, encouraging advanced training techniques for diffusion models and their applications.

ETHICS STATEMENT

Despite the remarkable contributions of recent advancement of image generation methods, several malicious usages have been reported such as deepfake and fake news. Since TADA is a simple and robust training method for diffusion models for unconditional image generation in limited training data, our method might be less likely to be used in malicious usages, compared to conditional generation such as text-to-image generation. Because our experiments are based on verified standard datasets, there are no ethical issues on data.

REPRODUCIBILITY STATEMENT

We ensure that TADA is reproducible with all hyper-parameters and design choices described in Appendix 6 and 7. We will make our code and checkpoints available at the time of release.

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

## APPENDIX

More details of our paper is explained throughout this appendix section.

Table 6: **Hyperparameters for 64×64 experiments.**

|  | FFHQ / AFHQ-v2 | Tab. 4 small | Tab. 4 big |
|---|---|---|---|
| $T$ | 1000 | 1000 | 1000 |
| $\beta_t$ | linear / cosine | linear | linear |
| Num. params (M) | 68 | 17 | 95 |
| Channels | 128 | 64 | 128 |
| Num. res blocks | 1 | 1 | 2 |
| Self-attn | 16, bottle | 16, bottle | 16, bottle |
| Heads channels | 64 | 64 | 64 |
| BigGAN block | yes | yes | yes |
| Dropout | 0.1 | 0.1 | 0.1 |
| Learning rate | 2e-5 | 2e-5 | 2e-5 |
| FP16 | - | - | - |
| EMA rate | 0.9999 | 0.9999 | 0.9999 |
| Batch size | 64 | 64 | 64 |
| Total steps | ≤ 100k | 50k | 50k |
| Num. images (M) | ≤ 6.4 | 3.2 | 2.5 |

Table 7: **Hyperparameters for 256×256 experiments.**

|  | FFHQ subset (1k, 2k, 5k) | AFHQ-v2 Cat (Fig. 6a) | Transfer (Tab. 3) |
|---|---|---|---|
| $T$ | 1000 | 1000 | 1000 |
| $\beta_t$ | linear | linear | linear |
| Num. params (M) | 138 | 138 | 94 |
| Channels | 128 | 128 | 128 |
| Num. res blocks | 2 | 2 | 1 |
| Self-attn | 32, 16, 8, bottle | 32, 16, 8, bottle | 16, bottle |
| Heads channels | 64 | 64 | 64 |
| BigGAN block | yes | yes | yes |
| Dropout | 0.1 | 0.1 | 0.1 |
| Learning rate | 2e-5 | 2e-5 | 2e-5 |
| FP16 | yes | yes | - |
| EMA rate | 0.9999 | 0.9999 | 0.9999 |
| Batch size | 16 | 16 | 16 |
| Total steps | 50k, 60k, 156250 | 156250 | 60k |
| Num. images (M) | 0.8, 0.96, 2.5 | 2.5 | 0.96 |

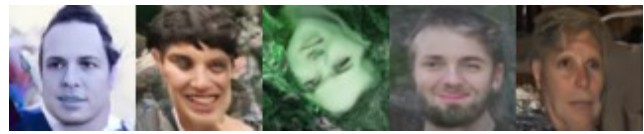

Figure 7: Example of generated images from a model trained with naive augmentation on FFHQ-5k 64 x 64 resolution.

## A    IMPLEMENTATION DETAILS

We employed the training objectives $L_{hybrid}$, as proposed by Nichol & Dhariwal (2021). Our implementations, including h-flip, AR, and TADA, used ADM (Dhariwal & Nichol, 2021) as the baseline architecture. For detailed configurations, including model architecture and training hyperparameters, please see Table 6 and Table 7.

## B    DISTRIBUTION SHIFT

Throughout this paper, we used the term *distribution shift* to describe a phenomenon where the generative model produces transformed samples, often reflecting the applied data augmentation. This terminology aligns with the usage in Zhao et al. (2020), and shares a similar meaning with the term *leaking* introduced in Karras et al. (2020).

Here, we provide the randomly sampled generated images in Figure 7.

## C    MORE RESULTS OF TOY EXPERIMENTS

### C.1    RESULTS ON DIFFERENT SETTINGS (TOY EXPERIMENT 1)

We show the result of toy experiment 1 (Section 3.2) on different settings. Figure 8a shows the LPIPS difference in cosine noise scheduling. Similar to the result in Figure 1a, the LPIPS difference in the *sensitive* region becomes significant, indicating that data augmentation has a comparably large effect in this region compared to the *rough* and *fine* timesteps. We further perform the same experiment on a different dataset. Figure 8b demonstrates the result on the AFHQ-v2 dataset, which shows a similar trend to the experiment done on FFHQ. In summary, this suggests that *sensitive* timesteps play a different role than other timestep intervals, regardless of different diffusion model settings or datasets.

### C.2    EFFECT OF SWAPPING IN DIFFERENT REGIONS (TOY EXPERIMENT 2)

This sections shows the result of toy experiment 2 (Section 3.2), by changing the timesteps for swapping region of $\hat{\epsilon}_{base} \rightarrow \hat{\epsilon}_{aug}$ and $\hat{\epsilon}_{aug} \rightarrow \hat{\epsilon}_{base}$. Figure 9a and 9b each illustrates a case when the model is swapped during $fine$ region, which starts from $r_{fine}$ (when log(SNR) = 0). The resulting samples of both figures did not present the change in the resultant distribution (e.g., change from the in-distribution to out-of-distribution and vice versa). This differs from Figure 1b, where the swapping occurred during *sensitive* timesteps. In other words, $\hat{\epsilon}_{aug}$ did not lead to producing distorted outputs and $\hat{\epsilon}_{base}$ did not result in-distribution samples. This indicates that when log(SNR) becomes larger than $r_{fine}$, both models cannot change the sampling trajectory as it becomes invertible after *sensitive* region, where no modifications are made to the context of the image.

We further provide results on AFHQ-v2 in Figure 10a, where the swapping occurs during *sensitive* region, thereby producing augmented-like output, similar to the result shown in Figure 1b. That is, $\hat{\epsilon}_{aug}$ leads to producing augmented-like outputs while $\hat{\epsilon}_{base}$ makes the output not deviate too far from the original data distribution. On the contrary, in Figure 10b, the sampling trajectory remains unchanged, similar to the result on FFHQ (Figure 9b). Overall, the results of these experiments imply that data augmentation plays a critical role during the *sensitive* timesteps.

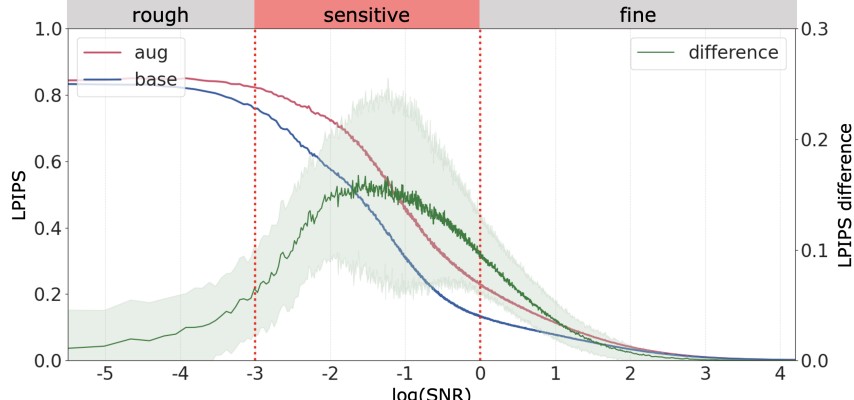

(a) **Different noise schedule**. Both models are trained on the FFHQ-5k dataset with a cosine noise schedule.

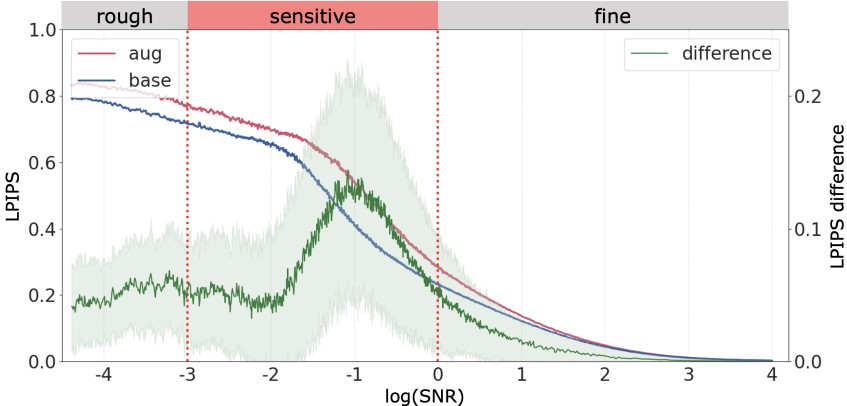

(b) **Different dataset**. Both models are trained on the AFHQ-v2 dataset.

Figure 8: **Results of toy experiments 1 on different configurations**.

## D   DETAILS OF TADA

### D.1   ON THE CHOICE OF $\delta$

To take advantage of data augmentation (e.g., generalization capability) in the *sensitive* timesteps, we chose to apply weak augmentations during this interval. However, to prevent distribution shift, we test the choice of $\delta$ by varying the augmentation strength. Specifically, we differ the degree of augmentation with $w_t = \delta \times max$, where $max$ denotes maximum augmentation strength defined in previous work (Karras et al., 2020). Figure 11 illustrates the mean faces when different values of $delta$ are used (multiplied to the maximum *scale* strength). As $\delta$ approaches 1, the size of the face becomes larger. We iterated this for every augmentation (e.g., scale, rotation) and found the safe value of $\delta$ as 0.1.

### D.2   CALCULATION OF $\kappa$

By default, we automatically compute the $\kappa$ so that the minimul $w_t$ is 0 before clipping. More specifically, given $r_{rough}$, $r_{fine}$ and $\delta$, we found the value of $\kappa$ that makes the minimum value of Equation 2 equal to 0. This can be implemented as follow:

```python
import numpy as np

def adapt(calibrated_snr, r_rough, r_fine, kappa=None, delta=0.1):
    logsnr = np.log10(calibrated_snr)
    if kappa is None:  # automatically compute the kappa parameter.
        kappa = 4.0*delta / (r_rough - r_fine)**2
```

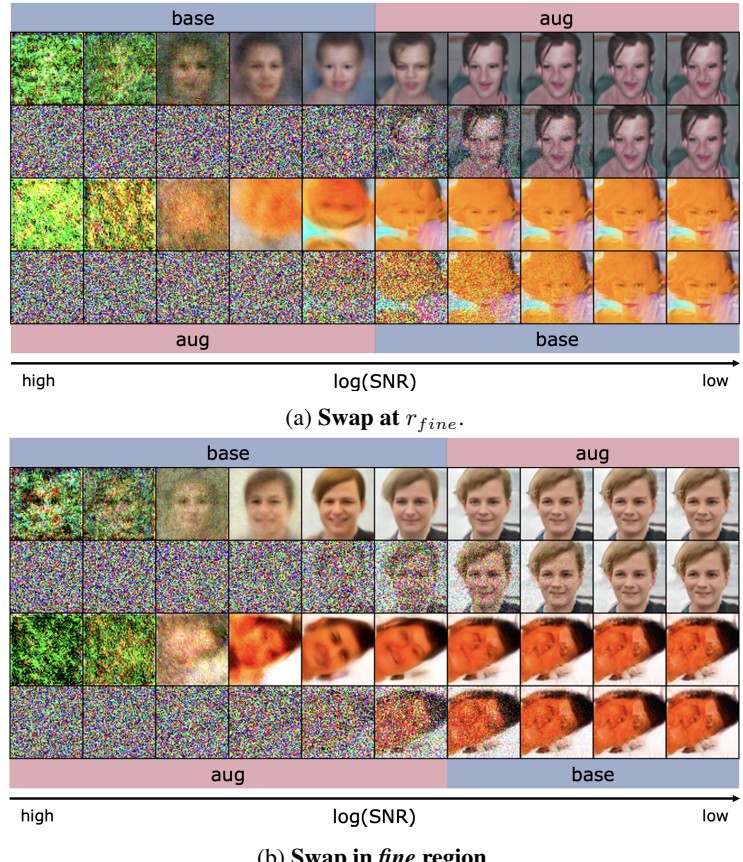

(a) **Swap at $r_{fine}$.**

(b) **Swap in *fine* region.**

Figure 9: **Results of toy experiment 2 on different regions**. Both models are trained on the FFHQ-5k dataset.

```
7     w = kappa * (logsnr-r_rough) * (logsnr-r_fine) + delta
8     return np.clip(w, 0, 1)
```

### D.3 AUGMENTATION PIPELINE

We adopted the augmentation pipeline from ADA (Karras et al., 2020), although we did not utilize all of its components. Specifically, we chose not to use *pixel blitting* as it falls within the category of *geometric* augmentations. We also excluded *lumaflip* because it is difficult to control its strength. Additionally, we opted not to employ image-space filtering (*filter*), as recent studies demonstrated that blurring (Hoogeboom & Salimans, 2023) and sharpening (Das et al., 2023) are directly related to diffusion models. Note that, the result of applying *filter* to our method can be found in Table 12. During the exploration stage of this paper, we conducted experiments with image-space *corruption* (i.e., cutout (DeVries & Taylor, 2017)) and obtained positive results. However, we decided not to include them for the sake of simplicity.

```
1  import numpy as np
2  transforms = [
3      scale, rotate_frac, aniso, translate_frac,
4      brightness, contrast, hue, saturation,
5  ]
6
7  def augment(image, snr, p=0.8, M=2, r_fine, r_rough, kappa, delta):
8      if np.random.rand(1) < p:
9          return image
10     w_t = adapt(snr, r_rough, r_fine, kappa, delta)
```

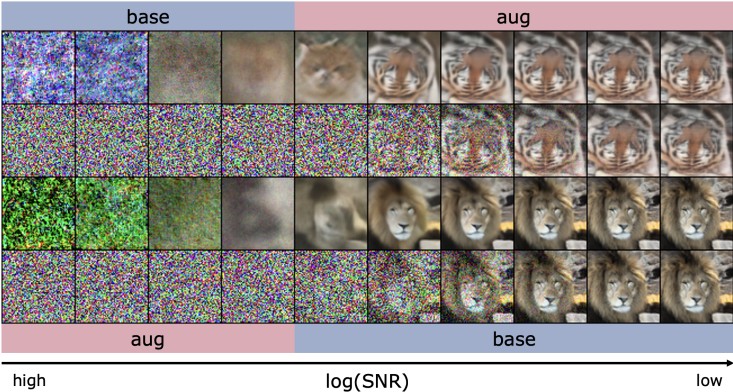

(a) **Swap in *sensitive* region**. Both models are trained on the AFHQ-v2 dataset.

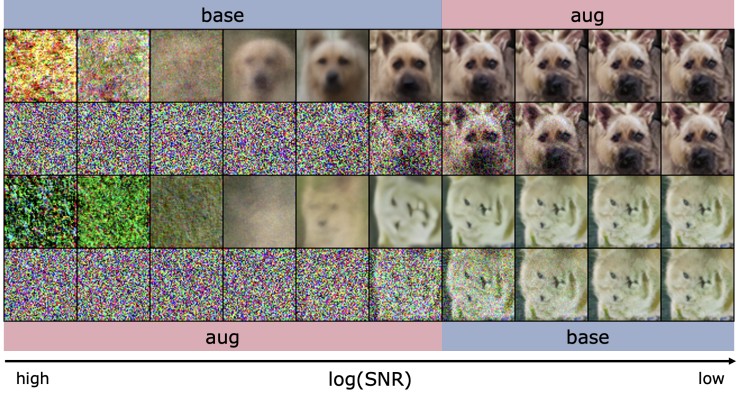

(b) **Swap in *fine* region**. Both models are trained on the AFHQ-v2 dataset.

Figure 10: **Results of toy experiment 2 on different regions**. Both models are trained on the AFHQ-v2 dataset.

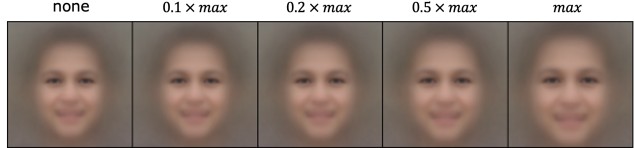

Figure 11: **On the Choice of $\delta$**. Example of *scale* augmentation.

```
11    num_apply = np.random.randint(1, M+1)
12
13    ops = np.random.choice(transforms, num_apply)
14    for op in ops:
15        image = op(image, w_t)
16    return image
```

# E    ABLATION ON *sensitive* REGION

We further evaluate our TADA by differing the values of $r_{rough}$ and $r_{fine}$. Table 8 and Table 9 show the results on FFHQ-5k and AFHQ-v2 Cat dataset, respectively. The best FID values of both tables are achieved when $[r_{rough}, r_{fine}] = [-2, 0]$, while the result of TADA $([-3, 0])$ achieved the second best. On AFHQ-v2 Cat, the resulting FID of $[-2, 1]$ shows the second-best values. We conjecture that the different trend in the AFHQ-v2 Cat is highly related to Figure 8b, as its peak in the LPIPS difference (green line) starts near -2, which is slightly different from the result

Table 8: **Ablation on FFHQ-5k. (64 x 64)** Results with 10k images and sampling steps = 250. Note that, $[r_{rough}, r_{fine}]$.

| Sensitive | FID ↓ | KID $(\times 10^3)$ ↓ | Precision | Recall |
|-----------|-------|------------------------|-----------|--------|
| $[-2, 1]$ | 14.28 | 8.92 | **0.78** | 0.30 |
| $[-2, 0]$ | **12.76** | **7.47** | 0.76 | **0.33** |
| $[-3, 1]$ | 15.94 | 10.73 | 0.77 | 0.31 |
| $[-3, 0]$ | 12.92 | 7.74 | 0.76 | **0.33** |
| h-flip | 14.84 | 9.30 | 0.78 | 0.30 |

Table 9: **Ablation on AFHQ-v2 Cat. (64x64)** Results with 10k images and sampling steps = 250. Note that, $[r_{rough}, r_{fine}]$.

| Sensitive | FID ↓ | KID $(\times 10^3)$ ↓ | Precision | Recall |
|-----------|-------|------------------------|-----------|--------|
| $[-2, 1]$ | 6.11 | 2.46 | 0.81 | 0.32 |
| $[-2, 0]$ | **5.80** | **2.29** | 0.80 | **0.35** |
| $[-3, 1]$ | 6.29 | 2.68 | **0.82** | 0.31 |
| $[-3, 0]$ | 8.33 | 3.87 | 0.77 | 0.27 |
| h-flip | 9.60 | 5.18 | 0.81 | 0.24 |

on *FFHQ* (Figure 1a). Although setting the *sensitive* regions narrower leads to a better FID score, we observed a trade-off between distribution shift and quality. Therefore, for devising a unified and simple augmentation solution, we safely set $[r_{rough}, r_{fine}]$ to $[-3, 0]$. Note that the values of $r_{rough}$ and $r_{fine}$ can be simply changed by the design choice.

# F   IN-DEPTH ANALYSIS ON DISTRIBUTION SHIFT

To further scrutinize into the relationship between timestep interval and distribution shift, we perform an in-depth analysis by seperating the whole timesteps into 10 intervals. Specifically, we seperate an interval each with 1 log(SNR) and train 10 distinct diffusion models, each trained with augmented data on specific interval. For example, the model is trained with augmented data during $-4 \leq$ log(SNR) $\leq -3$ and trained with original data distribution on the remaining intervals ($-inf \leq$ log(SNR) $\leq -4$ and $4 \leq$ log(SNR) $\leq inf$). We test with various types of augmentations (e.g., jigsaw (Jun et al., 2020) or cutmix (Yun et al., 2019)). Figure 14a and 14b each show the quantitative and qualitative analysis of distribution shift in each interval. For quantitative analysis, we count the number of unaligned samples among 10k generated samples by using our metric (Appendix F.1). For Figure 14b, we qualitatively evaluate the samples of each interval and mark whether the model generated augmented-like samples. Similar to the result of Figure 1a, the result of this section indicates that the distribution shift mainly occurs in *sensitive* interval.

## F.1   EVALUATION ON DISTRIBUTION SHIFT

We observed that FID does not penalize the emergence of distribution shift (Appendix F.2). While no previous studies have quantitatively assessed the distribution shift in generative samples, Karras et al. (2018; 2019) used qualitative visualizations to depict the significance of distribution shifts using the geometrically aligned dataset, such as FFHQ. By adopting this practice to design a quantitative metric, we measure the normalized number of generated samples with a noticeable difference in geometric alignment compared to the statistics of the FFHQ dataset (denoted as "Shift" (%)).

Specifically, We extract position vectors from facial landmarks, one for each of the center of the eyes ($c$), the x-axis ($x$), the y-axis ($y$), and the scale of the image ($s$). We then compute the mean and standard deviation of the position vectors extracted from 70,000 real FFHQ images and compare them to the vector values of a generated sample. By determining the threshold as $3 \times std$ of real images, any generated sample that falls outside this threshold is identified as a leaked sample. With this criterion, we count the number of leaked samples among 10k generated images for each timestep range.

| FFHQ-5k | [-3, -2] | [-2, -1] | [-1, 0] | [0, 1] | [1, 2] |
|---|---|---|---|---|---|
| v-flip | 12.99 | 12.32 | 12.24 | 11.89 | 13.40 |
| scale | 13.62 | 14.87 | 11.86 | 11.54 | 13.48 |

Figure 12: **FID values of each interval**.

| | [-4, -3] | [-3, -2] | [-2, -1] | [-1, 0] | [0, 1] |
|---|---|---|---|---|---|
| rotation | 50 | 1778 | 303 | 589 | 45 |
| transposition | 61 | 2238 | 611 | 80 | 39 |

Figure 13: **Number of *Shift* (%) in 256 × 256 resolution**.

### F.2 FID DOES NOT PENALIZE DISTRIBUTION SHIFT

Along this experiment, we discovered that FID itself does not penalize distribution shift. We measured the FID values on 10,000 generated samples of each interval, as shown in Figure 12, under the same setting as Figure 14a. Despite the generation of considerable number of unaligned dataset (Figure 14a), FID does not reflect the degree of distribution shift.

### F.3 SNR CALIBRATION

In this section, we measure the number of shifted samples on 256 x 256 resolutions. We perform the same experiment by training 10 models on the FFHQ-5k dataset of 256x256 resolutions, each with augmented training data on the corresponding interval. Figure 13 shows the result of this analysis, where the regions where distribution shift occurs are shifted by 1 log(SNR) in comparison with results on the 64x64 dataset (Figure 14a). Therefore, we perform SNR calibration as mentioned in Section 3.3.

## G EFFECTNESS OF TADA ON VARIOUS DATASETS

We further evaluate TADA on AFHQ-v2 Cat dataset. We compared the result with the model without any augmentation and the model trained on h-flip. TADA achieves the best results as shown in Table 10.

## H ON DESIGN OF CONDITIONAL INPUTS

Augmentation regularization requires additional conditional inputs for diffusion models. However, there is no standard implementation to on formulating these conditions. Therefore, we tested two publicly available implementations: the augmentation regularization (AR) (Karras et al., 2022) and DistAug (Jun et al., 2020), as used in (Kingma et al., 2021). We observed that differences in implementation details led to varying results. Table 11 shows the results. Note that, these methods require further design on augmentation labeling methods.

## I ADDITIONAL EXPERIMENTAL RESULTS

### I.1 QUALITATIVE RESULTS FOR TRANSFER LEARNING

In this experiment, we trained the pre-trained model on each domain of the AFHQ-v2 dataset. We obtained the pre-trained checkpoint provided by (Choi et al., 2022)[1]. See Figure 15, Figure 16, and Figure 17 for qualitative comparisons. Analogous to quantitative results in Table 3, our TADA with transfer learning provides superior results to AR, indicating our effectiveness in limited data settings.

### I.2 ABLATION STUDY ON AUGMENTATION POLICY

TADA employed various set of augmentations from (Karras et al., 2020), including *geometric* transformations (e.g., rotation, translation), *color*, and *filter*. In this experiment, we aim to understand the

---
[1]https://github.com/jychoi118/P2-weighting

| FFHQ-5k | (-inf, -4] | [-4, -3] | [-3, -2] | [-2, -1] | [-1, 0] | [0, 1] | [1, 2] | [2, 3] | [3, 4] | [4, inf) |
|---|---|---|---|---|---|---|---|---|---|---|
| v-flip | 29 | 32 | 49 | 1190 | 162 | 53 | 37 | 45 | 32 | 30 |
| scale | 40 | 39 | 39 | 64 | 494 | 227 | 66 | 43 | 46 | 45 |
| rotation | 37 | 40 | 38 | 248 | 588 | 87 | 47 | 31 | 48 | 47 |
| aniso | 41 | 45 | 55 | 52 | 226 | 71 | 46 | 55 | 35 | 53 |
| translation | 35 | 45 | 56 | 193 | 894 | 85 | 50 | 46 | 50 | 51 |

(a) Quantitative analysis on distribution shift.

| AFHQv2-Cat | (-inf, -4] | [-4, -3] | [-3, -2] | [-2, -1] | [-1, 0] | [0, 1] | [1, 2] | [2, 3] | [3, 4] | [4, inf) |
|---|---|---|---|---|---|---|---|---|---|---|
| rotation | x | x | x | △ | o | △ | x | x | x | x |
| transposition | x | x | △ | o | o | o | x | x | x | x |
| jigsaw | x | x | x | △ | o | △ | x | x | x | x |
| color swap | x | x | △ | o | o | △ | x | x | x | x |
| blur | x | x | x | x | △ | o | o | x | x | x |
| crop & resize | x | x | x | x | o | o | x | x | x | x |
| cutout | x | x | x | x | o | o | x | x | x | x |
| cutmix | x | x | x | x | o | o | x | x | x | x |

(b) Qualitative analysis on distribution shift.

Figure 14: **Analysis on distribution shift**. For quantitative analysis, we count a number of *Shift* (%) in the generated samples. For qualitative analysis, we manually investigate the generated images and assess the level of distribution shifts manually. ∘ implies the presence of noticeable distribution shifts, △ indicates the moderate amount of samples presenting distribution shifts, and × means no distribution shifts.

Table 10: **AFHQ-v2 cat (64x64).** Results with 10k images and sampling steps = 250.

| Methods | FID ↓ | KID $(\times 10^3)$ ↓ | Precision | Recall |
|---|---|---|---|---|
| none | 12.79 | 6.89 | **0.85** | 0.13 |
| h-flip | 9.60 | 5.18 | 0.81 | 0.24 |
| TADA | **8.33** | **3.87** | 0.77 | **0.27** |

impact of each augmentation by applying a single type (from the first to third row) and the combinations of two types (fourth row). Table 12 summarizes their contributions. In contrast to the findings of AR (Karras et al., 2022), who concluded that *color* augmentation yields no positive effects, our observations indicate that *color* augmentation actually leads to the most significant improvements.

Table 11: Augmentation regularization comparison. We tested two implementations on a 30k subset of FFHQ. All models were trained for 50,000 iterations. We report FID and KID using 10,000 images, which were generated using 50 sampling timesteps.

| Methods | FID ↓ | KID (×10³) ↓ |
|---|---|---|
| DistAug | 20.15 | 14.14 |
| AR | **18.91** | **12.66** |

Table 12: **Effects of various augmentation policies.** All models are trained on FFHQ 5k dataset (64 x 64). Values are average FID from 250 sampling timesteps.

| h-flip | rotation | scale | translation | aniso | color | filter | TADA |
|---|---|---|---|---|---|---|---|
| 14.82 | 13.89 (−0.29) | 13.97 (−0.88) | (+0.10) | 17.61 (+3.19) | (−1.50) | (+1.76) | 12.92 (−1.10) |

(a) Augmentation regularization             (b) `TADA`

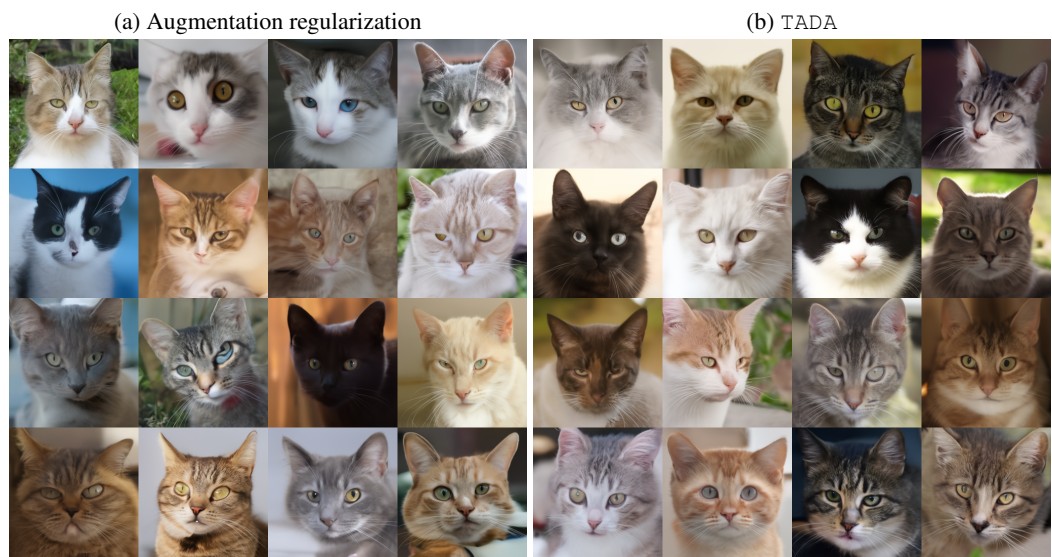

Figure 15: **Qualitative comparison.** The models are trained on AFHQ-v2 cat from the model pre-trained on FFHQ. The samples were generated with 50 sampling steps.

(a) Augmentation regularization             (b) `TADA`

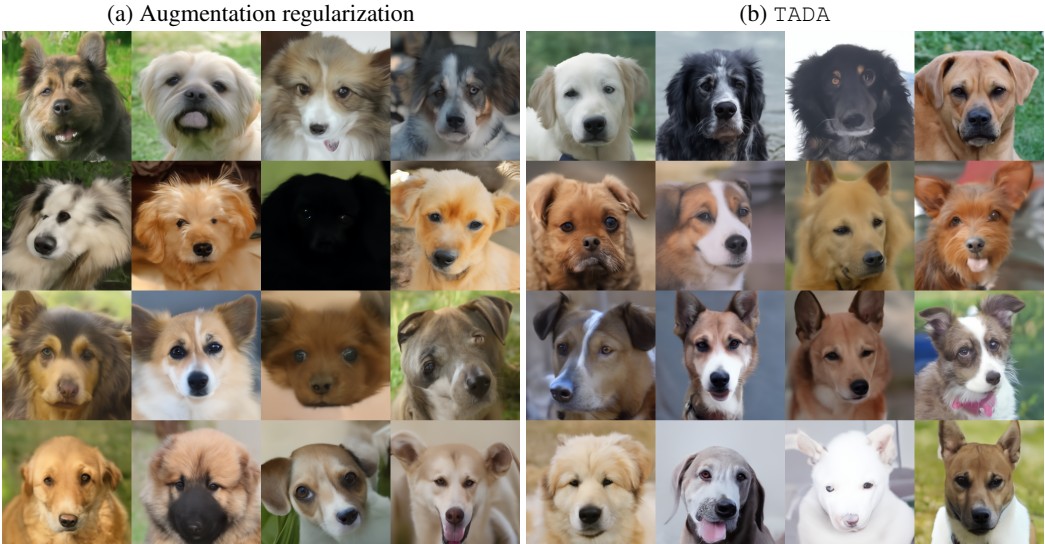

Figure 16: **Qualitative comparison.** The models are trained on AFHQ-v2 dog from the model pre-trained on FFHQ. The samples were generated with 50 sampling steps.

(a) Augmentation regularization  (b) TADA

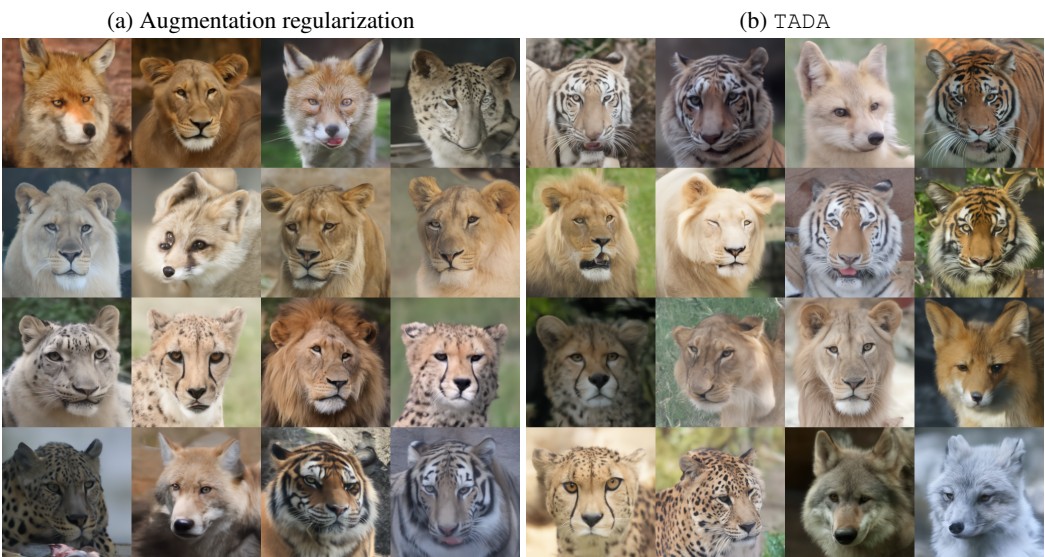

Figure 17: **Qualitative comparison.** The models are trained on AFHQ-v2 wild from the model pre-trained on FFHQ. The samples were generated with 50 sampling steps.

