# OpenReview forum: "TADA: Timestep-Aware Data Augmentation for Diffusion Models"
_ICLR.cc/2024/Conference — Submitted to ICLR 2024_

### Official Review · Reviewer_L1D2 · 2023-10-17

**Soundness:** 2 fair
**Presentation:** 3 good
**Contribution:** 2 fair
**Rating:** 3
**Confidence:** 4

**Summary:**

The authors proposed TADA, which uses SNR to control the strength of augmentation functions applied to the images in different timesteps when training diffusion models. The experimental results show the effectiveness of TADA on FFHQ and AHFQ-v2 datasets.

**Strengths:**

Although the idea of learning the augmentation strength is studied in image classification, using SNR as a measure to control the augmentation strength in generative diffusion models seems novel to me.

The paper is easy to follow.

**Weaknesses:**

(1) As TADA uses multiple types of augmentation, including h-flip, the improvement of TADA over the h-flip baseline may come from the use of other augmentations but not the proposed weighting scheme. The author should include more experiments to show the effectiveness of the time-aware weighting scheme, for example, comparing with a baseline with a fixed augmentation strength while keeping other configurations the same as TADA.

(2) TADA underperforms the AR baseline in various settings. The author should repeat the experiments multiple times and report the average and standard deviation of the results.

(3) The proposed timestep-aware weighting scheme is motivated based on the observation of two toy examples on 200 images from the FFHQ-5k dataset. Many of the parameters are selected manually: the augmentation probability, the choice of the augmentation functions, the SNR-sensitive region, and the maximum augmentation strength $\delta$. As observed from Table 9, the FID and KID increase if we use a sensitive region of [-3, 1] instead of [-3, 0]. Although the current setting is found to be effective on FFHQ and AHFA, it is unclear whether these pre-defined parameters need to be fine-tuned on any new datasets.

While it is acceptable that there are no solid theories supporting the timestep-aware mechanism, the method should be validated on sufficiently many scenarios, showing that the method is effective in general. I believe evaluating the method only on human faces (FFHQ) and animal faces (AHFQ) is not enough.

**Questions:**

The proposed TADA can only apply a timestep-aware global strength for all augmentations. The method cannot assign strength and probability values for individual augmentations, which may result in limited augmentation flexibility. In Table 8, applying only the color transformation leads to the best result. However, TADA cannot discover such an augmentation policy.

---

> ### Author Response · Authors · 2023-11-15
> **Response to Reviewer L1D2 (1)**
>
> We sincerely appreciate your time in reading the paper and constructive feedback to improve our paper.
>
> Please find below our detailed responses to your comments:
>
> **The proposed TADA can only apply a timestep-aware global strength for all augmentations. The method cannot assign strength and probability values for individual augmentations, which may result in limited augmentation flexibility.**
>
> We can assign different strengths and probability to individual augmentation. This can be checked on the provided source code. Specifically, a user can set the values of probability and adjust the strength as a hyperparameter. Furthermore, we will make our code publicly available and user-friendly.
>
> **In Table 8, applying only the color transformation leads to the best result. However, TADA cannot discover such an augmentation policy.**
>
> Our primary focus is to show the effectiveness of our method with a standard augmentation set, by adopting an augmentation set that is widely used in GANs literature. As mentioned in the limitations of our paper, the augmentation set we chose may not be optimal for diffusion models. Although not every augmentation has a positive effect, as shown in table 8, the augmentation overall gave benefits on the performance compared to the baseline. The optimal set can lead to additional improvement in performance. Thus, we believe that further research on finding the best set for diffusion models can be an interesting avenue.
>
> **The improvement may come from the use of other augmentations rather the proposed weighting scheme (comparison with fixed strength should be done)**
>
> We agree on your point that the benefit may come from the use of various augmentation sets. However, when we apply these various augmentations with fixed strength, the problem of distribution shift arises severely, as shown in the mean face of ‘naive’ augmentation in Figure 3. That is the reason we did not put the naive augmentation (the same set of augmentations with fixed strength) into the performance comparison table. Still, to provide clear results, we will add the FID scores of naive augmentation to our paper.
>
> **TADA underperforms the AR baseline. should repeat the experiments multiple times**
>
> We agree on your point that the experiment should be done multiple times with mean and std. We tested whether the results of our method vary with noticeable differences on one of the subsets where TADA underperforms AR. We measured the results three times and obtained a mean of 12.30 and a standard deviation of 0.09, which does not change any trend (lower than h-flip but higher than AR). Please note that AR used an augmentation probability of 0.12 or 0.15 depending on the dataset and we reported their best performance in comparison to our method.
>
> Throughout our experiments, our primary focus was not to assert TADA outperforms AR in all scenarios by manually setting parameters to surpass that of AR, but rather to emphasize its adaptability and ease of integration into various diffusion model settings while maintaining comparable performance to AR, with minimal tuning. We have tried to emphasize this point in our paper, but this point was not explicitly emphasized in our paper. We will revise our paper to make this clear.

---

> ### Author Response · Authors · 2023-11-15
> **Response to Reviewer L1D2 (2)**
>
> **Many of the parameters are selected manually. It is unclear whether these pre-defined parameters need to be fine-tuned on any new datasets.**
>
> For toy experiments, we reported the observations with 200 images by following the setting of P2 weighting [b]. During the design of this experiment, we confirmed that the number of chosen images does not affect the trend of this experiment. As mentioned in the paper, we mainly adopted augmentation sets from ADA [a], which is widely used in GANs literature. Therefore, we conducted our main experiments with a probability of 0.8 (that has been used as the maximum probability in ADA [a]) to demonstrate the feasibility of applying high augmentation probability with our proposed method. Since higher augmentation has been shown to be more effective in alleviating overfitting, we applied the highest probability possible. However, higher augmentation probability can lead to distribution shift, as reported in [a]. Therefore, being able to apply p=0.8 implies that our method allows usage of high probability and takes advantage of improved performance as well as alleviation of overfitting, without resulting in distribution shift. To further evaluate the effect of different probability parameters, we are currently performing probability sweeps with p=0.2, 0.4, and 0.6. We will report the results as soon as the training ends.
>
> Our proposed method has a primary focus on data augmentation techniques that do not cause distribution shifts. As pointed out, in Tables 8 and 9 of the appendix, performance can vary depending on the sensitive region. While the region boundaries that yield the best FID scores can be fluid, we consistently set the boundary to [-3, 0] across all datasets to prioritize robustness over maximizing performance. Indeed, performance could be further improved by fine-tuning these parameters. However, we emphasize that we avoided manually tuning the parameters to maximize performance and instead conducted our experiments using safe and moderate parameters.
>
> **Evaluations on only human faces are not enough**
>
> As we mainly focused on a data augmentation technique that does not lead to distribution shift, we utilized aligned datasets (e.g., subsets of FFHQ) that allow explicit evaluation of distribution shift in a qualitative manner, following ADA [a]. As reported in Tables 8 and 9 of the appendix, although the region boundaries that result in the best FID scores can be fluid, we consistently fixed the boundary to [-3, 0] across all datasets to prioritize robustness over performance maximization.
>
> However, we agree that showing effectiveness on an unaligned dataset can imply better generalizability, we are currently training on CIFAR-10. In addition, if time allows, we will also report on the LSUN dataset.
> We will make the results on these datasets available as soon as possible.
>
> Once again, we deeply appreciate the time you dedicated to reviewing our paper. We will incorporate your constructive feedback into revised versions of our paper.
> We respectfully ask if you consider raising the score. If you have any additional feedback that could further enhance our work, please share it with us at your convenience.
>
> [a] Karras, Tero, et al. "Training generative adversarial networks with limited data." Advances in Neural Information Processing Systems 33 (2020): 12104-12114.
>
> [b] Choi, Jooyoung, et al. "Perception prioritized training of diffusion models." Proceedings of the IEEE/CVF Conference on Computer Vision and Pattern Recognition. 2022.
>
> [c] Karras, Tero, et al. "Elucidating the design space of diffusion-based generative models." Advances in Neural Information Processing Systems 35 (2022): 26565-26577.

---

> ### Comment · Reviewer_L1D2 · 2023-11-22
>
> Thanks for the response. The authors provided more justifications for the experiment setup, e.g., the comparison with naive augmentations and the choices of the hyperparameters. The authors also emphasized their focus on the adaptability and the ease of integration. While these are favorable features of an augmentation scheme, it is essential to show the effectiveness of the method empirically and theoretically. Empirically, the current method sometimes underperforms the AR baseline and is evaluated on a limited choice of datasets. Theoretically, I agree with Reviewer xARD that the work lacks theoretical background explaining why the strength of the transformation at different timesteps helps suppress distribution shifts. Therefore I would like to keep my current score.

---

> > ### Author Response · Authors · 2023-11-22
> >
> > Thank you for your valuable feedback. We appreciate letting us know your concerns to our work. We will try to incorporate theoretical backgrounds in the future.

---

### Official Review · Reviewer_xARD · 2023-10-24

**Soundness:** 1 poor
**Presentation:** 1 poor
**Contribution:** 2 fair
**Rating:** 3
**Confidence:** 4

**Summary:**

This paper studies data augmentation in the training of diffusion models. Through two pre-experiments, the paper found that the level of impact of the data augmentation transformation on the output differs at each time step, especially in the interval called "sensitive time-steps," where the inverse diffusion process of the diffusion model significantly alters the input image. Based on this finding, the paper proposes a method to adjust the strengths of the data augmentation at each time step. Experiments show that the proposed method partially outperforms the existing baseline on FFHQ and AFHQ-v2. On the other hand, there is a lack of discussion on how the change in transformation strengths at each time step affects learning data distribution. In addition, the experiments are not extensive in evaluation datasets and diffusion model formulations, raising questions about the generality of the proposed method.

**Strengths:**

+ The paper is the first to investigate the impact of data augmentation at each time step in training a diffusion model.
+ Based on experimental observations, the paper proposes a heuristic data augmentation method that adjusts the strengths of the transformation at each time step.

**Weaknesses:**

- While the observations by Toy Examples 1 and 2 provided by the paper are interesting, the paper fails to explain why the distribution shifts arise in the diffusion model from these results (-> **Q1** and **Q2**).
- The theoretical foundation for the proposed method is weak. It is not clear why the proposed method solves the problems of distribution shift and overfitting (-> **Q3**).
- The experiments are limited in terms of the variations of datasets and the diffusion models. This limits the impact of the paper's claim and is inconsistent with the claims made in the Introduction (-> **Q4** and **Q5**).
- The experimental setup is unfair. The proposed method introduces color transformations that are not used in the baseline (AR), making it impossible to evaluate the proposed method with the experimental results. Also, in some of the experimental evaluations (Table 4 and Fig. 4), there is no comparison with the AR and the presentation is selective (-> **Q6** and **Q7**).
- The evaluation of the distribution shift in Fig. 3 is qualitative and subjective, and not convincing.
- The quality of the presentation with figures and tables is poor. Fig. 1(a), 4, and 5 cannot be accurately understood by a colorblind person like me.

**Questions:**

**Q1.** Why do the results in Fig. 1 indicate a distribution shift at sensitive time steps? To begin with, the definition of "distribution shift" here is vague and difficult to understand intuitively. According to the protocol in Sec. 3.2, the experiment in Toy Example 1 was performed by applying a data augmentation $T$ to an input image $x_0$ to generate $T(x_0)$, generating $x_t$ by adding noise, and applying an inverse diffusion step. Finally, the LPIPS between $\hat{x}_0$ and the original image $x_0$ are measured. This can evaluate the denoising performance of each step of the diffusion model, but it is unclear what LPIPS($x_0$, $\hat{x}_0$) means. Since $T(x_0)$ differs from $x_0$, we do not expect the inverse diffusion process to reconstruct $x_t$ into $x_0$. Therefore, it is unlikely that it makes sense to compare them. Also, the paper should provide more explanations about what it means theoretically to observe this LPIPS gap across data augmentations.

**Q2.** What does Toy Example 2 imply? The paper should specify what it wants to investigate by replacing $\hat{\epsilon}$. At least, I did not understand how the findings from this experiment are connected to the proposed method.

**Q3.** What is the theoretical background of the proposed method and what is the expected semantic behavior as a data augmentation? Why does the strength of the transformation at each time step help to suppress distribution shifts and overfitting?

**Q4.** Why does the paper experiment only with the facial datasets, i.e., FFHQ and AFHQ-v2? How about the performance of the proposed method on CIFAR-10 and ImageNet, which are used in EDM [a]? In the introduction, it is claimed that the proposed method is "applicable across various diffusion model settings", but since it is evaluated on fewer datasets than previous studies like EDM, the current results are not sufficient to claim the applicability.

**Q5.** Is the proposed method effective for formulations other than ADM? For example, how about using latent diffusion [b]? The paper should show that $r_\text{rough}$ and $r_\text{fine}$ are robust across the model formulations.

**Q6.** Why does the proposed method add transformations that are not used in AR? The paper states in section 4.1 that "we incorporate color transformations, which were not used in EDM". We could not compare the proposed method and AR without using shared transformations for the data augmentation because the effects of adding transformations cannot be separated from the effects of the proposed method.

**Q7.** Why are the AR results not listed in Table 4 and Fig. 4? Similarly, naïve data augmentation baselines that do not use the proposed method but use the same data augmentation transformations should be added to all experiments. The lack of these evaluations damages the significance of the paper.

[a] Karras, Tero, et al. "Elucidating the design space of diffusion-based generative models." Advances in Neural Information Processing Systems 35 (2022): 26565-26577.

[b] Rombach, Robin, et al. "High-resolution image synthesis with latent diffusion models." Proceedings of the IEEE/CVF conference on computer vision and pattern recognition. 2022.

---

> ### Author Response · Authors · 2023-11-15
> **Response to Reviewer xARD (1)**
>
> We appreciate your constructive comments and suggestions, and they are really helpful for us to improve our paper. We will carefully incorporate them into our paper.
> Furthermore, we apologize for inconvenience due to some figures that were not carefully made with appropriate colors. We have revised the corresponding figures (Fig.1(a), 4 and 5).  Please find below our detailed responses to your comments:
>
> **Q1. Why do the results in Fig. 1 indicate a distribution shift at sensitive timesteps?**
>
> For each model, we measured the perceptual similarity between the predicted $\hat{x}_0$ and the original image $x_0$ at each timestep of the reverse process. We then analyzed the two models' difference in the perceptual similarity.
>
> The baseline model, which was trained on the original dataset without data augmentation, is highly likely to produce in-distribution samples. Therefore, the perceptual similarity between the predicted $\hat{x}_0$ and the original image $x_0$ from this model can be thought of as the lower limit at each timestep. We then compare this with the augmented model. Since this model was trained with an augmented dataset, the generated samples will encompass both the in-distribution and out-of-distribution samples. This can be confirmed when the perceptual similarity between $\hat{x}_0$ and $x_0$ shows a noticeably higher perceptual difference if the predicted sample is an out-of-distribution sample.
>
> In summary, the difference in perceptual similarity between the two models is an indication of the degree to which the augmented model is producing out-of-distribution samples. The larger the difference, the more likely it is that the augmented model is producing samples that are not representative of the original dataset. The results in Fig. 1 align with this explanation and suggest that data augmentation can cause diffusion models to produce out-of-distribution samples, especially at sensitive timesteps.
>
> **Q2. What does Toy Example 2 imply?**
>
> Toy experiment 2 illustrates that exchanging the reverse process between the baseline and augmented models during the intermediate timesteps can modify the sampling trajectory, ultimately determining whether the generated samples end up within the augmented data distribution, even when they initially originated from the original data distribution, and vice versa. This highlights the influence of sensitive timestep on the sample distribution.
>
> The observation that exchanging the trajectory at other timesteps had no impact on the generated data distribution further strengthens the result of toy experiment 1 that the middle timesteps are the most susceptible to data augmentation (Figures 8 and 9). This implies that the model's ultimate generation to adhere to the original or augmented data distribution is predominantly made during these middle timesteps.

---

> ### Author Response · Authors · 2023-11-15
> **Response to Reviewer xARD (2)**
>
> **Q3. What is the theoretical background of the proposed method and what is the expected semantic behavior as a data augmentation? Why does the strength of the transformation at each time step help to suppress distribution shifts and overfitting?**
>
> We acknowledge that our data augmentation technique does not have a solid theoretical basis, and we view this as a limitation of our paper. Meanwhile, our paper is built upon our empirical findings that distribution shift occurs due to data augmentation and it appears in specific timestep intervals. To address this, we devised a method that adjusts the augmentation strength based on the timesteps. The reason that we view that the weak strength helps to suppress distribution shifts is that, we keep x_t of each timestep to not deviate too far and let the model learn how to construct the actual contents of an image, similar to findings from P2 weighting [a]. Effect on overfitting is not exclusively attributed to our proposed method (suppressing the augmentation strength) but can also result from the use of data augmentation itself [b]. The point we wanted to convey through our effectiveness on overfitting is to show that although we weaken the augmentation strength along specific timesteps, we still maintain the effectiveness on alleviating overfitting.
>
> As our method is built upon empirical evidence, we apologize that at this point, we cannot exactly provide answers with profound theoretical proofs. We will  endeavor to perform a deeper theoretical analysis on our future research. Still, we would like to emphasize the importance of this empirical study since we believe that empirical research can provide valuable insights and practical benefits, regardless of formal theoretical understanding. Indeed, a lot of theoretical studies are inspired and informed by empirical observations. One of the widely-used studies is CutMix [c], despite lacking rigorous theoretical investigations, has been utilized by many practitioners and following studies. CutMix has provided motivation for many theoretical studies such as [d, e]. We hope that empirical study is regarded as a valuable and essential form of research.
>
> [a] Choi, Jooyoung, et al. "Perception prioritized training of diffusion models." Proceedings of the IEEE/CVF Conference on Computer Vision and Pattern Recognition. 2022.
>
> [b] Karras, Tero, et al. "Training generative adversarial networks with limited data." Advances in Neural Information Processing Systems 33 (2020): 12104-12114.
>
> [c] Yun, Sangdoo, et al. "CutMix: Regularization strategy to train strong classifiers with localizable features." Proceedings of the IEEE/CVF International Conference on Computer Vision. 2019.
>
> [d] Park, Chanwoo, Sangdoo Yun, and Sanghyuk Chun. "A unified analysis of mixed sample data augmentation: A loss function perspective." Advances in Neural Information Processing Systems 35 (2022): 35504-35518.
>
> [e] Oh, Junsoo, and Chulhee Yun. "Provable benefit of mixup for finding optimal decision boundaries." International Conference on Machine Learning. PMLR, 2023.
>
> **Q4. Why does the paper experiment only with the facial datasets, i.e., FFHQ and AFHQ-v2?**
>
> As we mainly focused on a data augmentation technique that does not lead to a distribution shift, we utilized aligned datasets (e.g., subsets of FFHQ) that allow explicit evaluation of distribution shift in a qualitative manner, following ADA [b]. As reported in Tables 8 and 9 of the appendix, although the region boundaries that result in the best FID scores can be fluid, we consistently fixed the boundary to [-3, 0] across all datasets to prioritize robustness over performance maximization.
>
> However, we agree that showing effectiveness on an unaligned dataset can imply better generalizability, we are currently training on CIFAR-10. In addition, if time allows, we will also report on the LSUN dataset.
> We will make the results on these datasets available as soon as possible.

---

> ### Author Response · Authors · 2023-11-15
> **Response to Reviewer xARD (3)**
>
> **Q5. Is the proposed method effective for formulations other than ADM? For example, how about using latent diffusion?**
>
> A recent study [d] found out that latent diffusion has different timestep-wise task affinity compared to ADM (that we mainly used). Therefore, to apply our TADA to latent diffusion models, the boundary regions (r_rough and r_fine) should be adjusted.
>
> However, the majority of diffusion models that are currently used and studied are based on the same objective and architecture with variations on hyperparameters [a, b, c]. Therefore, we believe that our method can be seamlessly integrated into various diffusion models. To confirm this, we are currently training under VP [b]  and will provide the corresponding results.
>
> [a] Song, Yang, et al. "Score-based generative modeling through stochastic differential equations." arXiv preprint arXiv:2011.13456 (2020).
>
> [b] Karras, Tero, et al. "Elucidating the design space of diffusion-based generative models." Advances in Neural Information Processing Systems 35 (2022): 26565-26577.
>
> [c] Kingma, Diederik, et al. "Variational diffusion models." Advances in neural information processing systems 34 (2021): 21696-21707.
>
> **Q6. Why does the proposed method add transformations that are not used in AR?**
>
> We agree on your point that we employed color transformations that were not used in AR. This was because we tried to perform a fair comparison by presenting the best performance of AR methods, adhering to the specific augmentations that were found to be effective for EDM (color transformations were not used as authors of EDM mentioned that they found out color corruptions were rather harmful (Section F.2 of EDM [a])).
>
> Still, we agree on your point that the exact augmentation sets used for AR and TADA differ. Therefore, we are currently training TADA on the EDM augmentation set and training EDM on the augmentations that we used for TADA. We will report the results as soon as we obtain the corresponding results.
>
> [a] Karras, Tero, et al. "Elucidating the design space of diffusion-based generative models." Advances in Neural Information Processing Systems 35 (2022): 26565-26577.
>
> **Q7. Why are the AR results not listed in Table 4 and Fig. 4?**
>
> The purpose of Table 4 was to demonstrate the generalizability of TADA to various settings. Therefore, we performed a simple comparison with the most frequently used data augmentation for diffusion models (h-flip). Similarly, Figure 4 aimed to illustrate the effectiveness of TADA in addressing overfitting (albeit potentially trivial due to the well-known ability of data augmentations to mitigate overfitting), rather than directly comparing it to other data augmentation methods.
>
> Throughout our experiments, our primary focus was not to assert that TADA outperforms AR in all scenarios, but rather to emphasize its adaptability and ease of integration into various diffusion model settings, while maintaining comparable performance to AR. We have tried to emphasize this point in our paper, but this point was not explicitly emphasized in our paper. We will revise our paper to make this clear.
>
> Once again, we deeply appreciate the time you dedicated to reviewing our paper. We will incorporate your constructive feedback into revised versions of our paper. We respectfully ask if you consider raising the score. If you have any additional feedback that could further enhance our work, please share it with us at your convenience.

---

> ### Comment · Reviewer_xARD · 2023-11-22
>
> Thank you for the detailed response.
>
> **For the response to Q1**
>
> > In summary, the difference in perceptual similarity between the two models is an indication of the degree to which the augmented model is producing out-of-distribution samples.
>
> I'm less convinced by this interpretation because the meaning of the noise per time step for two diffusion models trained on different datasets is not necessarily the same. But, I could understand the argument of the paper with this reply. Thank you.
>
> **For the response to Q3**
>
> > Still, we would like to emphasize the importance of this empirical study since we believe that empirical research can provide valuable insights and practical benefits, regardless of formal theoretical understanding.
>
> My comment was not meant to say that there must always be a formal theoretical discussion. I agree that empirical research based on intuitive insights has practical importance and can motivate theoretical research. However, even with such an empirical approach, I believe that the presentation of a hypothesis and its empirical evidence is essential to a scientific paper. In that sense, I acknowledge that the paper did indeed present the hypothesis that there is a time step that is sensitive to domain shifts and attempted to demonstrate the validity of this hypothesis through empirical experiments. However, I still believe that the evidence and the influence of the hypothesis are not adequately discussed in the current experimental evaluation.
>
> **For the response to Q4 and Q6**
>
> I look forward to additional experimental results. Thank you for your efforts.

---

> > ### Author Response · Authors · 2023-11-22
> >
> > Thank you for your response. We appreciate your insights and comments, and we will try to incorporate them to improve the paper.
> >
> > Due to limited computational resources, we were unable to run many experiments.
> > We have tested our model on two additional datasets: LSUN church 64x64 and AFHQv2 Cat/Dog/Wild 256x256.
> >
> > * LSUN Church 64x64
> > |  | 1k        | 2k        | 5k        | 10k      | 30k      |
> > |--------:|-----------|-----------|-----------|----------|----------|
> > | h-flip | **21.06** | 32.38     | 14.28     | 8.48     | **6.67** |
> > | TADA   | 28.02     | **29.33** | **12.16** | **7.92** | 7.15     |
> >
> > * AFHQv2 256x256
> > |    | Cat       | Dog       | Wild     |
> > |--------:|-----------|-----------|----------|
> > | h-flip | **12.85** | 22.50     | 10.20    |
> > | tada   | 13.18     | **18.19** | **8.83** |
> >
> > **Q6**
> > We also trained AR with color transformation on FFHQ-5k 64x64, and the result was 12.47. This is almost the same as the result without color transformation (12.95).
> >
> > If you have any additional comments or feedbacks, we would appreciate further discussions.

---

### Official Review · Reviewer_zkiV · 2023-10-26

**Soundness:** 3 good
**Presentation:** 3 good
**Contribution:** 3 good
**Rating:** 6
**Confidence:** 2

**Summary:**

This paper presents a new data augmentation technique for the generative models using diffusion models. Data augmentation of generative models causes a distribution shift, which leads to the generation of unintended out-of-distribution samples. To avoid the distribution shift, the proposed method uses augmented data in early and last time steps in the reverse process. To justify this approach, this paper empirically reveals that only the middle time steps, which are called sensitive time steps, cause a distribution shift.
Experiments demonstrate that the proposed augmentation method outperforms the augmentation regularization method and 50% horizontal flip.

**Strengths:**

- This paper is well-written, and experiments are well-designed to present the claim of this paper.
- The empirical analyses about sensitive time steps are interesting and important. I think the existence of sensitive time steps will have an impact on future data enhancement techniques for diffusion models. I am not an expert in diffusion models, and it might not be very surprising that the process in the middle time step is sensitive. Even so, two toy experiments carefully confirm this phenomenon.
- This paper evaluates the proposed method from various perspectives.
For example, this paper evaluates the proposed method on different scales of the model, with/without transfer learning, and on high-resolution data. An ablation study is also contained, and I think the proposed method is sufficiently evaluated.

**Weaknesses:**

- It is difficult for the reader to reproduce the experiments in this paper alone. The proposed method is not very clearly presented because this paper does not contain pseudo codes or an explanation of the procedure of the whole proposed method. The specific data augmentation methods used in the proposed method are left to [Karras et al. (2020)] and not explained.
Thus, the paper by itself does not tell the readers how the strength of data augmentation $w_t$ works.
- This paper seems to evaluate the proposed method on only a limited data setting. Since data augmentation for discriminative models tends to improve performance when using full datasets besides limited data size settings, I would like to see the performance of the proposed method when a dataset has a sufficiently large size. If the proposed method does not improve the performance in this situation, it is the limitation of the proposed method.

**Questions:**

- Does the proposed method work well if a dataset has a sufficiently large size?
- Where can the readers understand the whole procedure of the proposed method? I think the readers have to carefully read several points e.g., Augmentation pipeline in Section 4.1, besides Section 3 to understand the proposed method, which is a bit burdensome.

---

> ### Author Response · Authors · 2023-11-15
> **Response to Reviewer zkiV**
>
> We sincerely appreciate your time in reading the paper, and the strengths that you found in our work. Please find below our detailed responses to your comments:
>
> **Does the proposed method work well if a dataset has a sufficiently large size?**
>
> As data augmentation is primarily employed in scenarios with limited data, our focus has been on evaluating its effectiveness in this context. Consistent with prior studies that demonstrate the benefits of data augmentation for small datasets, we presented results for datasets up to a size of 30k, following DiffAug [a] and ADA [b].
> However, we acknowledge the value of evaluating performance on the full FFHQ dataset to assess effectiveness and generalizability even under a large dataset. Accordingly, we are currently training TADA on the full FFHQ dataset and will provide the results as soon as the training is complete.
>
> **Where can the readers understand the whole procedure of the proposed method?**
>
> We sincerely apologize for any confusion regarding the proposed method’s procedure. To enhance clarity and reproducibility, we will provide more detailed information about the augmentation policy. Additionally, we have added the pseudo-code in the Appendix D.3 section to further explain our method’s implementation.
>
> Once again, we deeply appreciate the time you dedicated to reviewing our paper. We will incorporate your constructive feedback into revised versions of our paper.
>
> [a] Zhao, Shengyu, et al. "Differentiable augmentation for data-efficient gan training." Advances in neural information processing systems 33 (2020): 7559-7570.
>
> [b] Karras, Tero, et al. "Training generative adversarial networks with limited data." Advances in neural information processing systems 33 (2020): 12104-12114.

---

> > ### Comment · Reviewer_zkiV · 2023-11-22
> > **Thank you for the feedback**
> >
> > Thank you for the feedback. I think the claims of the paper and the proposed method are important results. However, I am not very familiar with the diffusion models and do not know the impact of the experimental results. Therefore, I keep my score and wait for discussion with other reviewers.
> >
> > > As data augmentation is primarily employed in scenarios with limited data, our focus has been on evaluating its effectiveness in this context.
> >
> > I understand the motivation, and the aim of my question was to clarify the limitation. I look forward to seeing the results of the full data.
> >
> > > Additionally, we have added the pseudo-code in the Appendix D.3 section to further explain our method’s implementation.
> >
> > The code in Appendix D.3 is unclear, and it is difficult to understand the role of each line; it relies on python and numpy code, and the relationship between the transforms and w_t is unclear. The detail explanation about transforms would be more reader-friendly even though they seen to be explained in ADA (Karras et al., 2020).

---

> > > ### Author Response · Authors · 2023-11-22
> > >
> > > Thank you for your response. We have tested our model using the full FFHQ-70k dataset and acquired the following results:
> > >
> > > | h-flip | tada |
> > > |--------|-------|
> > > | 15.22   | 16.67 |
> > >
> > > Consistent with the findings of ADA (Karras et al., 2020), our experiments indicate that data augmentation does not yield significant performance gains, implying that it may not be a crucial factor when training on a large and diverse dataset.
> > >
> > > We apologize for the ambiguity of our previous pseudo-code. We will try to make our code more reader-friendly.

---

### Official Review · Reviewer_o4Pj · 2023-11-08

**Soundness:** 3 good
**Presentation:** 3 good
**Contribution:** 3 good
**Rating:** 8
**Confidence:** 2

**Summary:**

The paper presents an interesting and novel approach to data augmentation in diffusion models by introducing a timestep-aware augmentation strategy. The authors identify a significant issue in the standard application of data augmentation to diffusion models, specifically the distribution shifts that occur and are tied to certain timesteps within the generative process. The proposed TADA approach is a logical and promising solution to this issue.

**Strengths:**

**Originality:** The paper addresses a gap in the literature by focusing on diffusion models where timestep-related distribution shifts during data augmentation have been understudied.

**Relevance:** Given the increasing importance of diffusion models in various generative tasks, the paper's focus is timely and relevant to the community.

**Technical Depth:** The paper appears to offer a comprehensive study and a well-thought-out augmentation strategy that is sensitive to the unique requirements of diffusion models.

**Empirical Evidence:** The authors provide experimental results that suggest their method is robust across different diffusion model settings and datasets, which is commendable.

**Weaknesses:**

**Clarity and Depth of Experiments:** While the experiments are diverse, the review is based on a limited preview, and a more thorough assessment of the experimental design and results is necessary.

**Comparison with Baselines:** The paper would benefit from a more detailed comparison with existing data augmentation methods, particularly in how the results translate to qualitative improvements in the generated samples.

**Impact and Significance:** The practical impact of the proposed method, while promising, is not fully established. The paper could strengthen its argument by demonstrating clear use cases where its method significantly outperforms existing techniques.

**Questions:**

**Methodology Clarification:**

Could you provide more detail on how the augmentation strength is adjusted during different timesteps? Is there a theoretical framework or empirical evidence that supports the chosen method?

**Experimental Design:**

In your experiments, did you explore the impact of the timestep-aware augmentation on the convergence speed and stability of the training process for diffusion models?
How does the proposed augmentation method affect the computational resources required for training and inference compared to traditional augmentation techniques?

**Baseline Comparisons:**

How does TADA compare with other state-of-the-art data augmentation methods in terms of qualitative and quantitative results?
Could you provide insights or metrics on how the proposed method improves the robustness and generalization of diffusion models in scenarios with limited data?

**Limitations and Challenges:**

What are the potential limitations or drawbacks of your proposed method?
Are there specific types of diffusion models or data domains where TADA might be less effective?

**Scalability and Generalization:**

How scalable is the TADA method with respect to model size and dataset complexity?
Have you tested the method on out-of-distribution samples or on datasets with significant class imbalances?

---

> ### Author Response · Authors · 2023-11-15
> **Response to reviewer o4Pj (1)**
>
> Thank you for your time and helpful feedback! We are really glad that you found our paper intriguing, comprehensive, and original. Please find below our detailed responses to your comments:
>
> **Methodology Clarification**
>
> As illustrated in Figure 2, augmentation strength (w_t) can be expressed as a quadratic curve across all timesteps. Specifically, since SNR can be determined as a function of timestep, we put the SNR value into Equation 3 to compute the augmentation strength w_t. Consequently, we adjust the augmentation strength by multiplying w_t to the maximum augmentation strength applied to training data. We newly provided an additional pseudo code describing the whole procedure in Appendix D.3.
>
> While the strength of the data augmentation can be kept constant across all timesteps, toy experiments suggest that diffusion models trained with augmented training data become more susceptible to producing unintended transformed samples during the intermediate timesteps. Based on this empirical evidence, we designed an augmentation strength schedule that suppresses the augmentation during the intermediate timesteps.
> The rationale behind our choice of a quadratic function is twofold:
>
> * Simplicity: The quadratic function provides a straightforward and intuitive approach to controlling the augmentation strength across timesteps. This simplicity makes it easy for users to adjust the augmentation intensity of a sensitive region without the need for complex hyperparameter tuning.
>
> * Smooth Transition: The quadratic function facilitates a seamless transition between strong and weak augmentation depending on the timestep axis. Toy experiments have demonstrated that data augmentation applied to sensitive timestep can significantly impact the generated samples. However, precisely identifying these sensitive regions is challenging. Therefore, we opted for a smooth function that ensures weak augmentation around sensitive timesteps while still enabling data augmentation to be applied to a substantial portion of the timesteps during training.
>
> This allows us to attain a balance between simplicity and effectiveness, making it a reasonable choice for the proposed method.
>
> We apologize for any ambiguity in the paper. We will revise the manuscript to improve the clarity of the methodology.
>
> **Experimental Design**
>
> Our proposed framework exhibits improved convergence speed and stability compared to the baseline (h-flip) and alleviates overfitting. As shown in Figure 4, FID scores computed every 10k training iterations indicate that our framework converges slightly faster than the baseline models. Additionally, our framework effectively alleviates overfitting, as evidenced by the lower FID scores at later stages of training.
>
> Employing our framework does not introduce any additional computational overhead during the training stage. Modern deep learning frameworks, such as PyTorch, execute data processing operations on the CPU, while the actual training process (forward and backward propagation) is performed on the GPU. Since our modifications are focused on the data processing stage, which is typically CPU-bound, they do not impact the training time.

---

> ### Author Response · Authors · 2023-11-15
> **Response to Reviewer o4Pj (2)**
>
> **Baseline Comparisons**
>
> While extensive research has been conducted on data augmentation for generative adversarial networks (GANs), limited research has explored this for auto-regressive and diffusion models. Specifically, no prior studies have investigated data augmentation for diffusion models, with only a rudimentary method introduced in EDM, leaving no other model for comparison. We believe this gap in research presents a significant opportunity for further exploration and advancement of data augmentation techniques for diffusion models.
>
> The reason that our method improves the robustness and generalization of diffusion models is not solely attributed to our proposed method but can also result from the use of data augmentation itself. Our method's primary impact lies in alleviating distribution shifts by adaptively adjusting the augmentation strength during specific timesteps. This strategy involves suppressing augmentation steps in regions where the diffusion model is susceptible to distribution shift, effectively mitigating this issue. Simultaneously, it allows for the application of diverse augmentation techniques with high augmentation probability, leveraging the benefits of data augmentation to promote better generalizability.
>
> **Limitations and Challenges**
>
> As we mentioned in the discussion section of the paper, limitation lies in the lack of a profound theoretical basis. The majority of diffusion models that are currently used and studied are based on the same objective and architecture with variations on hyperparameters [a, b, c]. Therefore, we believe TADA can be seamlessly integrated into various diffusion models. To confirm this, we are currently training under VP [a]  and will provide the results under this setting.
>
> Meanwhile, a recent study [d] found that latent diffusion has different timestep-wise task affinity compared to ADM (that we mainly used). Therefore, to apply our TADA to latent diffusion models, the boundary regions should be adjusted.
>
> Furthermore, we believe that our method is applicable across wide datasets with limited data settings. However, as we mentioned in the discussion of our paper, it can be less effective when an abundant amount of data is available.
>
> **Scalability and Generalization**
>
> We varied the model size (i.e., the number of parameters), and found that our method showed noticeable performance improvement in smaller model sizes (Table 4). This implies that TADA offers huge benefits when large computational resources and abundant training data are not available.
> In addition to the results on various sizes of the FFHQ dataset, we demonstrated our scalability on the AFHQ dataset. To further show the scalability of our method, we are currently training on CIFAR10. We will report the results as soon as the training is completed.
> We have not yet tested our method on out-of-distribution samples or class imbalance datasets. We appreciate your feedback on this point and hope that we can incorporate this aspect into our further research.
>
> Once again, we deeply appreciate the time you dedicated to reviewing our paper. We will incorporate your constructive feedback into revised versions of our paper.
>
> [a] Song, Yang, et al. "Score-based generative modeling through stochastic differential equations." arXiv preprint arXiv:2011.13456 (2020).
>
> [b] Karras, Tero, et al. "Elucidating the design space of diffusion-based generative models." Advances in Neural Information Processing Systems 35 (2022): 26565-26577.
>
> [c] Kingma, Diederik, et al. "Variational diffusion models." Advances in neural information processing systems 34 (2021): 21696-21707.
>
> [d] Go, Hyojun, et al. "Addressing Negative Transfer in Diffusion Models." arXiv preprint arXiv:2306.00354 (2023).

---

### Author Response · Authors · 2023-11-15
**General Response**

We sincerely thank every reviewer for valuable and constructive feedback on our paper. We are particularly grateful for the recognition of our work’s novelty (o4Pj, xARD, L1D2), clarity of presentation (zkiV, L1D2), and compelling experimental evidence (o4Pj, zkiV, xARD).

Our primary objective was to introduce a straightforward yet effective data augmentation technique that can be effortlessly integrated into the data processing pipeline. A key aspect we wish to emphasize is that simple data augmentation can enhance the performance of diffusion models while achieving comparable results to existing augmentation regularization methods that demand additional effort and training parameters on formulating conditioning input.

To demonstrate the generalizability of our proposed method, we refrained from optimizing the hyperparameters ($p$ and $M$) or augmentation pipeline. We focused on showing that TADA improves the performance of diffusion models over h-flip, the most commonly used data augmentation for training diffusion models, without requiring hyperparameter optimization for augmentation. We tested TADA on various datasets (including 5 subsets of FFHQ, AFHQ, and CIFAR-10), different noise schedules, and sampling steps. We demonstrated that our method is applicable across a variety of settings and is not sensitive or less effective in certain conditions. This user-friendly data augmentation pipeline is effective when trained from scratch and can be easily integrated with transfer learning settings. We hope our pipeline can be widely used to further enhance the generalization performance of diffusion models.

We acknowledge that TADA is grounded in empirical observations rather than theoretical derivation. While we hope future research can address this with deeper analysis and theoretical proofs, we would like to emphasize the importance of empirical study. Empirical research can provide valuable insights and practical benefits, even in the absence of formal theoretical understanding. Moreover, many theoretical studies are inspired and informed by empirical observations. One notable example is CutMix [a], which is not based on rigorous theory but has been utilized by many practitioners or subsequent studies. Furthermore, CutMix has fueled theoretical studies [b, c]. In this regard, we believe that empirical study should be considered an essential component of scientific inquiry.

To this end, we sincerely hope that our work will stimulate active research in the exploration and development of data augmentation techniques for diffusion models, as like the extensive research conducted for GANs, which is currently lacking for diffusion models.

We have provided detailed responses to each reviewer’s comments and we would be immensely grateful if the reviewer could review our responses and share any further feedback.

[a] Yun, Sangdoo, et al. "CutMix: Regularization strategy to train strong classifiers with localizable features." Proceedings of the IEEE/CVF International Conference on Computer Vision. 2019.

[b] Park, Chanwoo, Sangdoo Yun, and Sanghyuk Chun. "A unified analysis of mixed sample data augmentation: A loss function perspective." Advances in Neural Information Processing Systems 35 (2022): 35504-35518.

[c] Oh, Junsoo, and Chulhee Yun. "Provable benefit of mixup for finding optimal decision boundaries." International Conference on Machine Learning. PMLR, 2023.

---

> ### Author Response · Authors · 2023-11-15
> **Paper revision**
>
> Dear reviewers and AC.
>
> We have updated our manuscripts. In this version, we have made minor updates (e.g., fixing typos) and edited a number of figures for better visibility.
> We will consistently revise our paper for better flow and readability, with results on additional experiments.
>
> Sincerely,
> Authors.

---

### Meta-Review · Area_Chair_JfYc · 2023-12-06

**Metareview:**

2x R, 1x BA, and 1x A. This paper proposes a tilmestep-aware augmentation strategy in diffusion models. The reviewers agree on the important topic and clear writing, while are concerned about the missing theoretical foundation and insights, and unconvincing experiments (insufficient baselines, insufficient testing data, unfair setup, and suboptimal performance). The rebuttal did not fully address their concerns. The AC leans not to accept this submission.

**Justification For Why Not Higher Score:**

N/A

**Justification For Why Not Lower Score:**

N/A

---

### Decision · Program_Chairs · 2024-01-16

Reject